# Deep Global-sense Hard-negative Discriminative Generation Hashing for Cross-modal Retrieval

Kun Cheng[1], Qibing Qin[†2], Wenfeng Zhang[3], Lei Huang[4], and Jie Nie[4]

[1]Qufu Normal University, Rizhao, China, yuebai@qfnu.edu.cn
[2]Weifang University, Weifang, China, Qibing@wfu.edu.cn
[3]Chongqing Normal University, Chongqing, China, itzhangwf@cqnu.edu.cn
[4]Ocean University of China, Qingdao, China, {huangl, niejie}@ouc.edu.cn

## Abstract

Hard negative generation (HNG) provides valuable signals for deep learning, but existing methods mostly rely on local correlations while neglecting the global geometry of the embedding space. This limitation often leads to weak discrimination, particularly in cross-modal hashing, which learns compact binary codes. We propose Deep Global-sense Hard-negative Discriminative Generation Hashing (DGHDGH), a framework that constructs a structured graph with dual-iterative message propagation to capture global correlations, and then performs difficulty-adaptive, channel-wise interpolation to synthesize semantically consistent hard negatives aligned with global Hamming geometry. Our approach yields more informative negatives, sharpens semantic boundaries in the Hamming co-space, and substantially enhances cross-modal retrieval. Experiments on multiple benchmarks consistently demonstrate improvements in retrieval accuracy, verifying the discriminative advantages brought by global-sense HNG in cross-modal hashing. Related code and data are available at https://github.com/QinLab-WFU/DGHDGH.

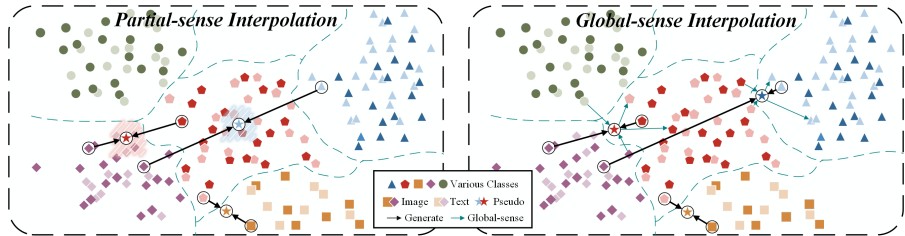

Figure 1: Traditional generation methods only interpolate based on the correlation between single anchor-negative pairs, which damages the global distribution relationship of heterogeneous samples in the embedding co-space. Through the interpolation of hard negative samples with global awareness of sample correlation, the generated samples are controlled to avoid violating the feature distribution in the embedding space, which makes the co-space more discriminative.

## 1 Introduction

Deep Cross-modal Hashing Retrieval (DCHR) aims to learn deep hash functions that project heterogeneous samples into compact hash codes within a shared Hamming embedding space, such that semantically similar heterogeneous samples are assigned similar codes, and dissimilar ones are mapped to distinct codes Hu et al. (2023); Zhang et al. (2023); Liu et al. (2019). This property transforms cross-modal retrieval into a simple and efficient hash-based search Luo et al. (2023); Han et al. (2026); Li et al. (2025b); Qin et al. (2025).

---

[†]Corresponding Author.

To enhance discriminability, one effective strategy is to provide more informative signals during training Rubinstein et al. (1997); Cakir et al. (2019). Informative learning methods can generally be categorized into mining-based and generation-based approaches Wu et al. (2017); Peng et al. (2024). Currently, hard negative mining is the most widely used strategy Wang et al. (2025); Xuan et al. (2020a). Difficult samples provide stronger adversarial signals, yield larger gradient updates, and force the model to learn more discriminative representations Kalantidis et al. (2020); Cheng et al. (2026); Xia et al. (2022). However, mining is constrained by the scarcity of naturally occurring hard samples within each mini-batch, limiting its effectiveness during training Zheng et al. (2019); Zhang et al. (2022); Vasudeva et al. (2021). Hard negative generation (HNG) addresses this issue by synthesizing more challenging samples, typically through linear interpolation of existing negatives, thereby enriching informative learning Peng et al. (2024); Yang et al. (2023).

Despite these advances, most existing works focus solely on local neighborhoods for negative interpolation, failing to capture the global geometric structure across diverse classes, an issue particularly pronounced in the cross-modal co-space. As shown in Fig. 1, traditional interpolation strategies select distant negative samples and create harder negatives based solely on anchor–negative correlations. For example, when selecting blue text embeddings as negatives for a purple image anchor, the interpolated sample may mistakenly fall into the red category distribution. This failure arises because local interpolation ignores the influence of other categories and the overall global distribution. Consequently, generated samples often intrude into non-original semantic regions, thereby weakening discriminability.

To overcome this issue, we propose learning global sample correlations and explicitly modeling inter-class relationships during generation, enabling the synthesis of informative negatives with appropriate difficulty that respects the semantic manifold. Specifically, we introduce Deep Global-sense Hard-negative Discriminative Generation Hashing (DGHDGH), which performs Discriminative Global-sense Synthesis (DGS) guided by Relevance Global Propagation (RGP). In the RGP, we construct a structured graph where nodes store embeddings and edges encode pairwise relevance. Through iterative message propagation, each edge learns global-sense correlations. The DGS then uses these correlations to perform channel-wise adaptive interpolation, ensuring the generated samples remain semantically consistent. Unlike traditional methods that apply a single coefficient across all channels Ko & Gu (2020); Venkataramanan et al. (2022), our approach adapts difficulty per channel, with an additional self-paced mechanism to regulate generation hardness throughout training. Moreover, no extra generator network is required, improving adaptability and efficiency.

In summary, the main studies of this paper are listed as shown below.

- Firstly, we propose a novel DGHDGH framework, which is the first attempt, to the best of our knowledge, to introduce hard negative generation into cross-modal hashing. By learning global sample relevance and synthesizing hardness-adaptive negative samples, DGHDGH achieves more discriminative cross-modal retrieval.
- Secondly, we devised the RGP module, which uses graph neural networks to establish global heterogeneous sample correlation perception in order to determine the appropriate difficulty of synthetic negatives and enhance the semantic alignment of synthetic samples in the co-space.
- Thirdly, we designed the DGS module to flexibly generate channel-wise hardness adaptive negatives based on global relationships, thereby enhancing informative hash learning.
- Finally, extensive experiments on three benchmarks demonstrate that the proposed DGHDGH learns a discriminative Hamming co-space through informative hash learning with global-sense HNG, surpasses state-of-the-art methods in retrieval performance, and can serve as a plug-and-play module to enhance existing cross-modal hashing approaches.

## 2  RELATED WORKS

Deep Cross-modal Hashing Retrieval (DCHR) has been extensively studied for aligning heterogeneous modalities in a shared Hamming space Chen et al. (2023); Li et al. (2023). Early works primarily emphasized supervised semantic alignment, while more recent approaches introduced hierarchical structures, neighborhood-preserving mechanisms, or uncertainty estimation to enrich training signals Li et al. (2025c); Qin et al. (2024); Huo et al. (2024b). Despite these advances,

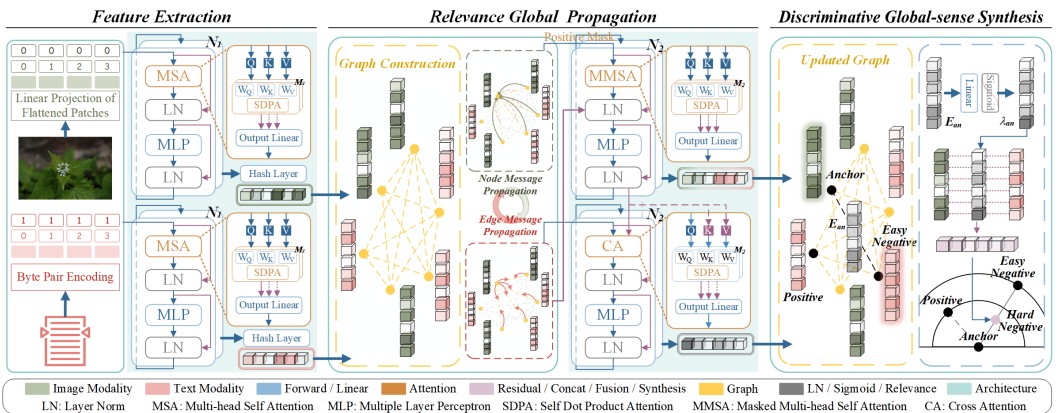

Figure 2: The schematic of our proposed DGHDGH framework. (1) We employ a dual transformer architecture with hash layers to extract hash codes from heterogeneous data synchronously. (2) RGP represents codes of the entire batch by a graph and introduces an iterative graph message propagation mechanism via another dual transformer that updates nodes and edges alternately. (3) DGS uses the learned global relevance to produce interpolation vectors for each anchor-negative pair to get a harder version with discrimination.

most methods still rely on fixed training pairs and lack mechanisms for generating informative hard negatives, which constrains their discriminative capability Duan et al. (2018); Zheng et al. (2019).

Existing approaches to informative learning can be broadly divided into two families. Mining-based methods explicitly select particular forms of samples to maximize the extracted information, such as Distance Weighted Sampling (DWS) Wu et al. (2017). Augmentation-based methods instead create additional supervision signals, including generator-based approaches such as GANs, interpolation-based strategies like Dense Anchor Sampling (DAS), and memory-based mechanisms like Cross Batch Memory (XBM) Cao et al. (2018b); Liu et al. (2022); Wang et al. (2020b).

While hard negatives play a crucial role in improving model discrimination, the effectiveness of hard negative mining is often limited by the number of available samples Bucher et al. (2016); Xuan et al. (2020a). Hard negative generation (HNG) has therefore emerged as a promising alternative. Most existing methods obtain relationships through interpolation or generate features via a separate generator, but they generally focus only on local correlations, which can distort semantic consistency. For example, Hardness-adaptive Deep Metric Learning (HDML) Zheng et al. (2019) synthesizes samples based on local neighborhoods, yet fails to align the generated negatives with the global geometry of the embedding space Peng et al. (2024). To address this limitation, we propose a novel HNG framework tailored for cross-modal hashing. Our method leverages global feature perception to generate hardness-adaptive negatives that better preserve semantic alignment across modalities, thereby enhancing discriminative retrieval. An extended discussion is provided in Appendix A.

## 3 METHODOLOGY

### 3.1 FEATURE EXTRACTION

A schematic of the proposed DGHDGH framework is shown in Fig. 2. Let $x^{\mathcal{I}}$ and $x^{\mathcal{T}}$ denote image and text modality samples from a multi-modal dataset $\mathcal{D} = {x_i^{\mathcal{I}}, x^{\mathcal{T}}i, l_i}^n i = 1$. Semantic features $h^{\mathcal{I}}$ and $h^{\mathcal{T}}$ of length $K$ are obtained through the hash functions $F^{\mathcal{I}}$ and $F^{\mathcal{T}}$. Here, $l_i \in {0, 1}^{N \times C}$ is the common multi-hot label vector for the $i$-th heterogeneous pair $(x_i^{\mathcal{I}}, x_i^{\mathcal{T}})$, where $N$ denotes the number of samples and $C$ the number of categories. To generate hash codes, we adopt Transformer-based feature extraction by employing dual Transformers for the image and text modalities. Each Transformer contains $N_1$ blocks followed by a hash layer. A block consists of a Multi-head Self-Attention (MSA) module with $M_1$ heads and a Multi-Layer Perceptron (MLP), separated by Layer Normalization (LN) and equipped with residual connections. The hash layer consists of an MLP followed by a $tanh$ activation. Since binary optimization is a prototypical NP-hard problem, the

$tanh$ function is used as a continuous relaxation strategy to learn binary-like codes during training.

$$\tilde{h}^* = tanh(\text{MLP}(z^{N_1*})) \in (-1,1)^{N \times K}, * \in \{\mathcal{I}, \mathcal{T}\} \quad (1)$$

During testing, the $sign$ function is leveraged to obtain binary codes:

$$h^* = sign(\tilde{h}^*) \in \{-1,1\}^{N \times K}, * \in \{\mathcal{I}, \mathcal{T}\} \quad (2)$$

In the following sections, we omit the superscripts $\mathcal{I}, \mathcal{T}$, when the modality distinction is not critical. where $z^{N_1}$ denotes the features learned by the $N_1$-th block. For the features $z^k$ learned in the $k$-th block, the update rule is:

$$z_i^{k+1} = \text{LN}\Big(\text{MLP}(z_i') + z_i'\Big), \quad \text{where } z_i' = \text{LN}\Big(\text{MSA}(z^k)_i + z_i^k\Big). \quad (3)$$

Through this process, semantic-preserving hash codes can be effectively learned. However, the resulting codes still suffer from insufficient discriminability. To address this, we introduce a global-sense hard negative generation method to enhance training informativeness, consisting of two modules: Relevance Global Propagation (RGP) and Discriminative Global-sense Synthesis (DGS).

## 3.2 RELEVANCE GLOBAL PROPAGATION

To effectively generate information-rich hard negatives, it is crucial to determine both their appropriate difficulty level and spatial distribution. Thus, when selecting interpolation points for each anchor–negative pair, their similarity should be evaluated relative to all other samples in the global batch context. To this end, we construct a structured graph to capture sample associations across the entire batch and employ a graph network to learn global correlations.

Initially, we assign the batch features $\tilde{h}$ into the structured graph $\mathcal{G} = (V, E)$ as nodes $V_i^k|k = 0 = \tilde{h}_i$. Edges $E$ represent pairwise correlations, initialized as $Eij^k|_{k=0} = \tilde{h}_i \odot \tilde{h}_j$. We maintain three graphs in parallel: image, text, and cross-modal. The first two take samples from their respective modalities, while the cross-modal graph contains all heterogeneous samples. We then introduce a graph transformer ($GT$) with $N_2$ blocks and $M_2$ heads for each block, to learn sample relationships globally via iterative message propagation. The three graphs share parameters and are jointly updated in $GT$, which helps narrow the cross-modal semantic gap and improves robustness. Message propagation adopts a dual-transformer architecture that updates nodes and edges separately. Unlike the synchronous feedforward dual Transformer in feature extraction, the node and edge Transformers here perform asynchronous alternating updates—first propagating node messages, then edge messages. This ordered procedure ensures that node information continuously informs subsequent edge updates, thereby improving the model's ability to capture and exploit global sample correlations.

For the node Transformer, we design a Masked Multi-head Self-Attention (MMSA) mechanism with a positive mask, which ensures that each node (treated as an anchor) interacts only with its negative samples. In MMSA, each node is treated as a query, and all corresponding negative samples are treated as keys and values. To prevent disproportionately high attention weights from weakening discrimination among subtle negative differences, positive samples are masked—especially heterogeneous identical samples in the cross-modal scenario. We further introduce edge-to-node interactions after MMSA, incorporating neighboring edge information into nodes to enrich representations and strengthen global context understanding. The main formula of the $k$-th node transformer block is shown as follows:

$$V_i^{k+1} = \text{LN}\Big(\text{MLP}(V_i') + V_i'\Big), \quad \text{where } V_i' = \text{LN}\Big(\text{MMSA}(V^k)_i + \sum_{j=1}^{\mathcal{B}} E_{ij}^k + V_i^k\Big). \quad (4)$$

For the edge Transformer, we introduce node-to-edge interactions via a Cross-Attention (CA) mechanism. Here, edge representations act as queries, while node representations serve as keys and values, allowing edges to integrate information from neighboring nodes. This allows edges to capture the relevance of their critical points from a global perspective and further adjust their attention trends, thereby enabling edges to adaptively balance the difficulty of synthesizing anchor-negative pairs. The formula of the $k$-th edge transformer block is shown as follows:

$$E_{ij}^{k+1} = \text{LN}\Big(\text{MLP}(E_{ij}') + E_{ij}'\Big), \quad \text{where } E_{ij}' = \text{LN}\Big(\text{CA}(E_{ij}^k, V_i^{k+1}, V_j^{k+1}) + E_{ij}^k\Big). \quad (5)$$

After $n_2$ iterations of message propagation, i.e., $n_2$ dual-transformer blocks, the edge information is expected to encode sufficient global correlations to enrich the information content of synthetic negatives.

A more formal discussion of the propagation behavior of RGP, including how it differs from classical smoothing-based graph operators, is provided in Appendix B.

## 3.3 DISCRIMINATIVE GLOBAL-SENSE SYNTHESIS

Based on the learned global sample relevance, we dynamically interpolate and fuse anchor–negative representations with channel-wise adaptivity, producing more informative negatives that enhance training and strengthen the discrimination of the embedding space. Initially, use the edges $E_{an}^{n_2}$ of each anchor-negative pair to obtain the corresponding interpolation vector $\lambda_{an}$:

$$\lambda_{an} = Sigmoid(FC(E_{an}^{n_2})) \tag{6}$$

where FC denotes a Fully Connected layer used for transformation, and the $Sigmoid$ function performs normalization. Thus, $\lambda_{an}$ can provide adaptive weights for channel-level embedding fusion in the corresponding anchor-negative pairs.

Unlike traditional interpolation methods that apply a single coefficient, we gradually increase training difficulty as the model converges. Therefore, we define the interpolation formula as follows:

$$\tilde{h}'_{an} = \begin{cases} (1-\eta)\tilde{h}_a + \eta\tilde{h}_n, & \text{if } d_{ap} < d_{an}, \\ \tilde{h}_n, & \text{otherwise.} \end{cases} \quad \text{where } \eta = \Big(d_{ap} + \lambda_{an}\tau(d_{an} - d_{ap})\Big)/d_{an} \tag{7}$$

where $\tau$ is introduced as a dynamic scaling factor for adjusting interpolation points, gradually increasing the difficulty of synthesizing negative samples during model training. We set $\tau = e^{-\frac{1}{l_{avg}}}$, where $l_{avg}$ is measured by the average loss from the previous epoch, reflecting the model's current learning performance. As the model gradually fits, $l_{avg}$ decreases, and $\tau$ gradually tightens the upper bound of the interpolation interval, increasing the difficulty of synthetic negative samples. $\lambda_{an}$ is responsible for generating appropriate deterministic values within the interpolation interval to achieve informative interpolation based on global propagation of correlations.

## 3.4 GENERATION OPTIMIZATION

To optimize the generation of difficult samples, we design multiple loss functions that guide the model toward the desired objectives from different perspectives. We expect the generated samples to have a higher similarity (difficulty) with the anchors while maintaining the original semantics, so we designed two losses: **Semantic Preservation loss** $\mathcal{L}_{sp}$ and **Interpolation Similarity loss** $\mathcal{L}_{is}$. To calculate the semantic preservation loss, we add an extra classification layer to the model. This layer is trained only on real samples and then used to classify synthetic samples, without backpropagating gradients from the synthetic inputs. The calculation formula of $\mathcal{L}_{sp}$ is as follows:

$$\mathcal{L}_{sp}(\tilde{h}'_{an}) = CE(CL(\tilde{h}'_{an}), l_n) \tag{8}$$

where CL denotes the classification layer, which is essentially an FC layer. CE is the $cross-entropy$ function, and $l_n$ is the original negative sample category.

For the $\mathcal{L}_{is}$, we directly use cosine similarity function to calculate:

$$\mathcal{L}_{is}(\tilde{h}'_{an}, \tilde{h}_a) = 1 - \frac{\tilde{h}'_{an} \odot \tilde{h}_a}{||\tilde{h}'_{an}|| \, ||\tilde{h}_a||} \tag{9}$$

To encourage diversity among synthetic negatives, the interpolation coefficients $\lambda$ should vary across pairs. Thus, for each anchor $a$, the standard deviation of all associated coefficients $\lambda_{a-}$ defines the **Coefficient Diversity loss** $\mathcal{L}_{cd}$:

$$\mathcal{L}_{cd}(\lambda_{a-}) = 1 - \sigma(\lambda_{a-}) \tag{10}$$

where $\sigma$ represents the standard deviation function ($std$).

The overall **Generation Optimization loss** $\mathcal{L}_{go}$ is defined as:

$$\mathcal{L}_{go} = \gamma_{is}l_{is} + \gamma_{sp}l_{sp} + \gamma_{cd}l_{cd} \tag{11}$$

where $\gamma_{is}$, $\gamma_{sp}$, and $\gamma_{id}$ are used to adjust the weights of the loss items. Through a comprehensive assessment of three aspects, we enhance the model's ability to generate informative negative samples, thereby enabling more discriminative hash learning.

### 3.5 Hash Learning

After obtaining diverse synthetic samples, we focus on strengthening discriminative hash learning while maintaining robustness. Since we designed a classification layer in the generation optimization section to evaluate the semantic consistency of synthetic samples, we need to add this layer after the hash layers and train it using the corresponding loss:

$$\mathcal{L}_{sp1} = CE(CL_1(\tilde{h}_i), l_i) \tag{12}$$

At the same time, in order to maintain semantic consistency in the graph neural network, we also pass the node representations after graph message propagation through a classification layer, so that the nodes continue to maintain their semantics while learning global information:

$$\mathcal{L}_{sp2} = CE(CL_2(V_i^{n_2}), l_i) \tag{13}$$

Note that these two classification layers differ from the modality-specific hash layers; instead, they share parameters across modalities, similar to $GT$, to enhance robustness. This is because we aim for the feature codes obtained from the hash layers to already eliminate modality differences, thereby allowing them to be directly applied during testing.

We adopt the standard triplet loss for hash learning, incorporating both real and synthetic hard negatives to verify the effectiveness. We first compute the triplet loss using real samples only:

$$\mathcal{L}_{\text{real}} = \mathcal{L}_{\text{tri}}(\tilde{h}^{\mathcal{I}}, \tilde{h}^{\mathcal{I}}) + \mathcal{L}_{\text{tri}}(\tilde{h}^{\mathcal{I}}, \tilde{h}^{\mathcal{T}}) + \mathcal{L}_{\text{tri}}(\tilde{h}^{\mathcal{T}}, \tilde{h}^{\mathcal{I}}) + \mathcal{L}_{\text{tri}}(\tilde{h}^{\mathcal{T}}, \tilde{h}^{\mathcal{T}}) \tag{14}$$

where $\mathcal{L}_{\text{tri}}$ represents the triplet loss function. We then introduce the synthetic hard negative samples generated by our DGS module to further strengthen the learning process. The enhanced triplet loss with synthetic negatives is defined as:

$$\mathcal{L}_{\text{syn}} = \mathcal{L}_{\text{tri}}(\tilde{h}^{\mathcal{I}}, \tilde{h}^{\mathcal{I}\mathcal{I}\prime}) + \mathcal{L}_{\text{tri}}(\tilde{h}^{\mathcal{I}}, \tilde{h}^{\mathcal{I}\mathcal{T}\prime}) + \mathcal{L}_{\text{tri}}(\tilde{h}^{\mathcal{T}}, \tilde{h}^{\mathcal{T}\mathcal{I}\prime}) + \mathcal{L}_{\text{tri}}(\tilde{h}^{\mathcal{T}}, \tilde{h}^{\mathcal{T}\mathcal{I}\prime}) \tag{15}$$

where $\tilde{h}^{\mathcal{I}\prime}, \tilde{h}^{\mathcal{I}\mathcal{T}\prime}, \tilde{h}^{\mathcal{T}\mathcal{I}\prime}, \tilde{h}^{\mathcal{T}\prime}$ represent the synthetic hard negative samples generated for the respective modality pairs, which are interpolated by Eq. 7. Among them, $\tilde{h}^{\mathcal{I}\mathcal{T}\prime}$ represents the synthetic samples with $\mathcal{I}$ as the anchors and $\mathcal{T}$ as the negatives.

The overall **hash learning loss** $\mathcal{L}_{hl}$ combines the real and synthetic triplet losses:

$$\mathcal{L}_{\text{hl}} = \mathcal{L}_{\text{real}} + \gamma_{\text{syn}}\mathcal{L}_{\text{syn}} \tag{16}$$

where $\gamma_{\text{syn}}$ is set to $1 - e^{\frac{1}{\mathcal{L}_{go}}}$. As $GT$ converges, it progressively increases the proportion of hard negatives to strengthen metric learning.

The overall training procedure alternates between $\mathcal{L}_{go}$ and $\mathcal{L}_{hl}$, ensuring that both sample generation and hash code learning are jointly improved throughout the training process. This coordinated optimization strategy enables our model to learn highly discriminative hash codes that effectively preserve semantic similarities across modalities.

## 4 Experiments

### 4.1 Benchmark Datasets & Baseline Methods

**MIRFLICKR-25K** contains 24,581 image-text pairs across 24 semantic categories from the Flickr website Huiskes & Lew (2008). **NUS-WIDE** was constructed by the National University of Singapore, contains 195,834 pairs, 21 classes Chua et al. (2009). **MS COCO** created by Microsoft, contains 122218 sample pairs from 80 categories Lin et al. (2014). In our experiments, those three datasets are split identically by randomly selecting 10,000 image-text pairs as the training set. Afterwards, 5000 pairs are chosen randomly as the query set and the remaining as the database.

To demonstrate the performance of our method comprehensively, we have chosen several typical deep cross-modal hashing methods to compare with our proposed DGHDGH framework, which include Two-step discrete hashing (TwDH)Tu et al. (2024), Deep Neighborhood-aware Proxy Hashing (DNPH)Huo et al. (2024a), Deep Neighborhood-preserving Hashing (DNpH)Qin et al. (2024), Deep Hierarchy-aware Proxy Hashing (DHaPH)Huo et al. (2024b), Bi-Direction Label-Guided Semantic Enhancement Hashing (BiLGSEH)Zhu et al. (2025), Deep Evidential Hashing (DECH) Li

Table 1: mAP@all results(%) of DGHDGH and baseline methods on three benchmark datasets w.r.t. four hash bits .

| Task | Method | Reference | MIRFLICKR-25K | | | | NUS-WIDE | | | | MS COCO | | | |
|---|---|---|---|---|---|---|---|---|---|---|---|---|---|---|
| | | | 16 | 32 | 64 | 128 | 16 | 32 | 64 | 128 | 16 | 32 | 64 | 128 |
| Image ↓ Text | TwDH | TMM'24 | 79.71 | 81.47 | 83.19 | 84.37 | 66.83 | 69.34 | 69.95 | 71.94 | 64.29 | 70.04 | 73.08 | 75.44 |
| | DNPH | TOMM'24 | 81.08 | 82.69 | 82.89 | 83.70 | 66.89 | 68.11 | 69.39 | 70.93 | 64.38 | 69.10 | 72.94 | 72.51 |
| | DNpH | TMM'24 | 84.23 | 85.52 | 85.88 | 86.29 | 69.21 | 70.22 | 70.71 | 71.58 | 67.27 | 69.03 | 68.60 | 68.74 |
| | DHaPH | TKDE'24 | 82.99 | 84.37 | 85.31 | 85.49 | 69.58 | 70.35 | 71.36 | 71.55 | **72.84** | 74.15 | 74.75 | 75.43 |
| | BiLGSEH | TCSVT'25 | 79.29 | 81.16 | 81.94 | 82.07 | **70.50** | 71.42 | 72.18 | 72.13 | 66.68 | 73.33 | 75.96 | 74.85 |
| | DECH | AAAI'25 | 79.61 | 83.96 | 83.83 | 84.43 | 66.13 | 71.61 | 71.55 | 72.41 | 63.73 | 64.35 | 66.44 | 68.49 |
| | DPBE | MM'25 | 80.82 | 83.27 | 85.12 | 85.90 | 62.46 | 64.51 | 68.35 | 71.14 | 62.56 | 64.77 | 69.26 | 72.61 |
| | DDBH | TCSVT'25 | 84.50 | 85.34 | 86.10 | 86.50 | 69.34 | 71.45 | 72.29 | 72.29 | 71.65 | 74.54 | 76.81 | 78.24 |
| | **DGHDGH** | **OURS** | **84.66** | **86.19** | **87.13** | **87.75** | 69.72 | **71.68** | **72.60** | **73.76** | 72.06 | **74.71** | **77.13** | **79.19** |
| Text ↓ Image | TwDH | TMM'24 | 77.80 | 80.01 | 81.96 | 82.96 | 67.06 | 71.02 | 71.37 | 72.60 | 65.68 | 70.92 | 74.45 | 76.11 |
| | DNPH | TOMM'24 | 80.15 | 81.76 | 81.66 | 82.32 | 68.71 | 69.94 | 71.82 | 71.91 | 64.68 | 70.12 | 73.88 | 72.98 |
| | DNpH | TMM'24 | 81.47 | 82.92 | 83.61 | 84.22 | 69.92 | 71.37 | 71.39 | 72.31 | 65.62 | 68.60 | 69.28 | 68.87 |
| | DHaPH | TKDE'24 | 81.48 | 81.65 | 82.29 | 82.79 | 68.78 | 70.54 | 69.98 | 70.42 | 69.35 | 70.69 | 71.54 | 71.87 |
| | BiLGSEH | TCSVT'25 | 80.48 | 82.41 | 83.43 | 83.47 | 70.27 | 70.89 | 72.02 | 73.24 | 68.96 | 73.16 | 75.43 | 74.72 |
| | DECH | AAAI'25 | 78.69 | 81.85 | 82.23 | 83.67 | 68.28 | **73.05** | 73.15 | 73.18 | 62.11 | 65.27 | 66.97 | 69.15 |
| | DPBE | MM'25 | 79.31 | 81.54 | 83.55 | 84.06 | 63.49 | 66.23 | 69.63 | 73.42 | 62.56 | 64.23 | 71.75 | 74.93 |
| | DDBH | TCSVT'25 | 82.45 | 83.18 | 83.90 | 84.33 | 70.23 | 72.11 | 73.25 | 73.53 | **71.67** | 73.94 | 75.95 | 77.07 |
| | **DGHDGH** | **OURS** | **83.03** | **84.21** | **85.09** | **85.74** | **70.75** | 72.64 | **73.75** | **74.64** | 71.16 | **74.69** | **77.41** | **79.59** |

The best and second-best performance are highlighted in boldface and underline.

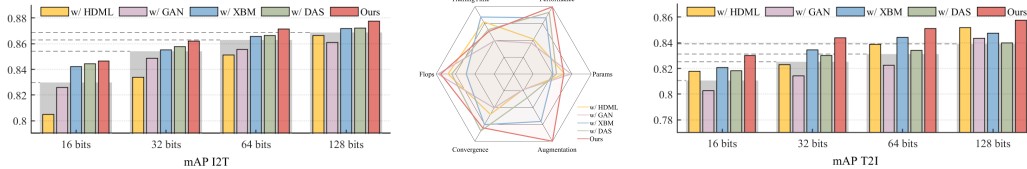

Figure 3: Performance comparison with augmentation-based methods on MIRFLICKR-25K, and use mining-based w/ DWS as the gray background.

et al. (2025c), Deep Probabilistic Binary Embedding (DPBE) Cheng et al. (2025) Deep Discriminative Boundary Hashing (DDBH)Qin et al. (2025). To ensure fairness, all frameworks adopt CLIP ViT-B/32 as the common backbone, and the experimental settings are kept the same except for the hyperparameters set in the original paper. Furthermore, different types of informative methods are picked, namely Distance-Weighted Sampling (w/ DWS) Wu et al. (2017), hashGAN (w/ GAN) Cao et al. (2018b), Hardness-adaptive Deep Metric Learning (w/ HDML) Zheng et al. (2019), and Densely-Anchor Sampling (w/ DAS) Liu et al. (2022).

## 4.2 EVALUATION METRICS & IMPLEMENTATION DETAILS

We evaluate cross-modal similarity search in two settings: Image-to-Text (I2T) and Text-to-Image (T2I). We primarily use mean Average Precision (mAP), which reflects both recall and precision, along with the Fisher ratio and $P@H \leq 2$ to evaluate model discriminability.

The initial parameters of the feature extraction module are referenced in Radford et al. (2021), where $N_1 = 12$, $M_1 = 8$. For parameter optimization, we utilize the Adam optimizer, where a learning rate of 0.001 and a weight decay of 0.01. We set the batch size as 128 and take the best performance in 100 epochs for all experiments. A detailed description can be found in Appendix C

## 4.3 PERFORMANCE COMPARISON

To rigorously verify the performance of our proposed DGHDGH, we report the comparison with baseline methods as shown in Tab. 1. By learning a discriminative Hamming co-space, our method achieves state-of-the-art performance results. Meanwhile, in order to comprehensively demonstrate the difference with previous information learning, we further compare DGHDGH with representative generation methods on MIRFLICKR-25K, as shown in Fig. 3, which are all based on the same baseline. Other metrics are also evaluated in a normalized radar plot, which are Flops, Parameters (reverse), Training times (reverse), convergence epochs (reverse), and information augmented. The metrics were all performed Max normalization, and the maximum value was taken as the largest or slightly larger constant. See more baseline experiments in Appendix D.1 and D.2.

Table 2: Fisher ratios (%) of DGHDGH and baseline methods w.r.t. four hash bits on three benchmark datasets, which are computed by randomly sampling 50, 100, 200, and 400 positive/negative pairs, each repeated five times with different seeds for stability.

| MIRFLICKR | 16 bits | | 32 bits | | 64 bits | | 128 bits | |
|---|---|---|---|---|---|---|---|---|
| | $I{\to}T$ | $T{\to}I$ | $I{\to}T$ | $T{\to}I$ | $I{\to}T$ | $T{\to}I$ | $I{\to}T$ | $T{\to}I$ |
| DHaPH | 90.15 ± 0.14 | 84.38 ± 0.19 | 104.54 ± 0.14 | 88.16 ± 0.07 | 109.22 ± 0.17 | 90.89 ± 0.21 | 108.28 ± 0.18 | 92.46 ± 0.08 |
| BiLGSEH | 69.81 ± 0.11 | 69.05 ± 0.06 | 76.43 ± 0.17 | 75.86 ± 0.06 | 80.42 ± 0.19 | 82.62 ± 0.07 | 81.36 ± 0.18 | 84.46 ± 0.08 |
| DECH | 81.13 ± 0.13 | 67.85 ± 0.06 | 96.38 ± 0.13 | 80.76 ± 0.15 | 95.41 ± 0.16 | 82.94 ± 0.10 | 91.55 ± 0.10 | 87.98 ± 0.13 |
| DDBH | **100.13 ± 0.12** | 84.51 ± 0.07 | 104.65 ± 0.13 | 84.51 ± 0.11 | 109.21 ± 0.26 | 90.89 ± 0.08 | **111.30 ± 0.10** | 92.47 ± 0.06 |
| **DGHDGH** | 97.57 ± 0.05 | **89.02 ± 0.14** | **105.44 ± 0.08** | **93.16 ± 0.11** | **111.17 ± 0.15** | **94.60 ± 0.05** | 110.17 ± 0.07 | **96.82 ± 0.15** |
| **NUS-WIDE** | | | | | | | | |
| DHaPH | 101.03 ± 0.13 | 98.86 ± 0.16 | 90.15 ± 0.14 | 90.05 ± 0.16 | 107.21 ± 0.15 | 104.04 ± 0.14 | 106.15 ± 0.13 | 103.22 ± 0.13 |
| BiLGSEH | 99.77 ± 0.13 | 101.87 ± 0.11 | 100.32 ± 0.21 | 102.42 ± 0.17 | 99.00 ± 0.18 | 103.24 ± 0.14 | 99.00 ± 0.18 | 103.21 ± 0.13 |
| DECH | 93.92 ± 0.14 | 97.85 ± 0.13 | 105.56 ± 0.14 | 108.89 ± 0.11 | 97.69 ± 0.15 | 105.49 ± 0.10 | 104.69 ± 0.13 | 109.24 ± 0.09 |
| DDBH | **110.09 ± 0.17** | **110.02 ± 0.17** | 112.72 ± 0.17 | 111.14 ± 0.15 | **115.86 ± 0.11** | 113.44 ± 0.14 | 116.49 ± 0.15 | 116.31 ± 0.15 |
| **DGHDGH** | 105.03 ± 0.20 | 104.68+0.10 | **112.89 ± 0.22** | **112.70 ± 0.12** | 113.27 ± 0.18 | **114.41 ± 0.07** | **117.58 ± 0.13** | **118.80 ± 0.09** |
| **MS COCO** | | | | | | | | |
| DHaPH | 116.94 ± 0.17 | 105.43 ± 0.07 | 113.73 ± 0.14 | 114.38 ± 0.11 | 117.05 ± 0.10 | 120.65 ± 0.07 | 120.58 ± 0.15 | 119.58 ± 0.09 |
| BiLGSEH | 90.76 ± 0.11 | 90.07 ± 0.12 | 99.46 ± 0.14 | 100.66 ± 0.14 | 105.93 ± 0.08 | 106.60 ± 0.08 | 106.24 ± 0.12 | 106.50 ± 0.09 |
| DECH | 88.55 ± 0.18 | 90.90 ± 0.05 | 94.46 ± 0.12 | 105.41 ± 0.11 | 95.77 ± 0.10 | 108.17 ± 0.10 | 98.15 ± 0.12 | 107.36 ± 0.11 |
| DDBH | **123.63 ± 0.17** | 124.29 ± 0.18 | 131.57 ± 0.15 | 123.37 ± 0.11 | **136.25 ± 0.21** | 124.64 ± 0.15 | **139.10 ± 0.20** | 126.46 ± 0.10 |
| **DGHDGH** | 120.63 ± 0.26 | **124.56 ± 0.11** | **128.36 ± 0.18** | **131.45 ± 0.21** | 128.17 ± 0.26 | **132.58 ± 0.12** | 133.20 ± 0.17 | **135.94 ± 0.14** |

The best and second-best performance are highlighted in boldface and underlined.

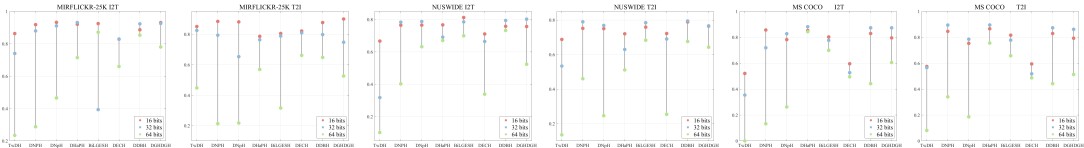

Figure 4: $P@H \le 2$ results of DGHDGH and baseline methods on three benchmark datasets.

## 4.4 DISCRIMINATIVE HASHING

We argue that introducing richer discriminative information during training facilitates more discriminative hash learning. To evaluate this, $P@H \le 2$ is utilized to demonstrates the compactness of the learned Hamming co-space. In Fig. 4, the experimental result measures how well each model pushes away hard negatives, validating the discriminative capability of our proposed method.

On the other hand, we assessed discrimination by comparing the Fisher ratio. As shown in Tab. 2, our method achieves higher Fisher ratios than all baselines. This indicates that the proposed global-sense hard negative generation leads to a more discriminative Hamming space, leading to tighter intra-class clusters and larger inter-class separations. It is worth noting that the two methods that performed well in DDBH, similarly emphasize discriminative properties.

## 4.5 SELF VALIDATION

To fairly judge the contributions of our modules, we conduct ablation studies to evaluate each component separately in Fig. 5. For w/o RGP, we directly use the initial edge computation as the interpolation source . For w/o DGS, we directly remove the generation phase. Furthermore, we validate two detailed operations in two modules. Furthermore, we validate two finer operations in the modules: removing Edge Message Fusion (w/o EMF) in RGP, and removing the Hardness-Adaptive Parameter (w/o HAP) in DGS. At the same time, we also investigate the three optimization objectives for generative embedding, i.e., $\mathcal{L}_{is}, \mathcal{L}_{sp}$ and $\mathcal{L}_{cd}$, and cross ablate them in Tab. 3. These three loss terms

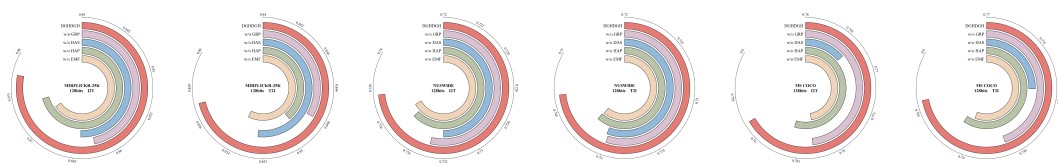

Figure 5: Ablation Study Result of DGHDGH on three benchmark datasets w.r.t. 128 bits.

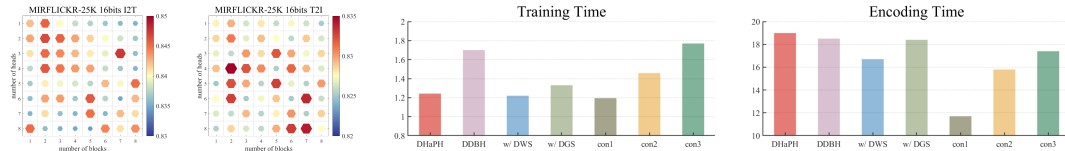

Figure 6: Parameter configuration and temporal effects Results on MIRFLICKR-25K w.r.t. 16 bits.

Table 3: Ablation Study Result of DGHDGH on MIRFLICKR-25K.

| Component | | | 16 bits | | 32 bits | | 64 bits | | 128 bits | | Avg. | |
|---|---|---|---|---|---|---|---|---|---|---|---|---|
| $l_{sm}$ | $l_{sp}$ | $l_{id}$ | $I{\to}T$ | $T{\to}I$ | $I{\to}T$ | $T{\to}I$ | $I{\to}T$ | $T{\to}I$ | $I{\to}T$ | $T{\to}I$ | $I{\to}T$ | $T{\to}I$ |
| | | | 79.82 | 78.15 | 81.45 | 79.63 | 82.21 | 80.57 | 83.06 | 81.32 | 81.64 | 80.04 |
| ✓ | | | 81.38 | 79.52 | 82.97 | 80.73 | 83.84 | 81.69 | 84.61 | 82.58 | 83.20 | 81.13 |
| | ✓ | | 82.15 | 80.37 | 83.82 | 81.64 | 84.59 | 82.81 | 85.33 | 83.45 | 83.97 | 82.07 |
| ✓ | | ✓ | 83.76 | 82.14 | 85.41 | 83.28 | 86.25 | 84.36 | 87.02 | 84.95 | 85.61 | 83.68 |
| | | ✓ | 83.92 | 82.35 | 85.63 | 83.51 | 86.47 | 84.59 | 87.21 | 85.17 | 85.81 | 83.90 |
| ✓ | ✓ | | 84.44 | 82.91 | 86.09 | 83.95 | 86.82 | 84.98 | 87.38 | 85.43 | 86.18 | 84.32 |
| | ✓ | ✓ | 82.84 | 81.06 | 84.55 | 82.42 | 85.38 | 83.67 | 86.12 | 84.23 | 84.72 | 82.85 |
| ✓ | ✓ | ✓ | **84.66** | **83.03** | **86.19** | **84.21** | **87.13** | **85.09** | **87.75** | **85.74** | **86.43** | **84.52** |

The best and second-best performance are highlighted in boldface and underlined.

optimize generated hard negatives in terms of interpolation similarity, semantic preservation, and parameter diversification, respectively. The figure demonstrates that optimizing the interpolation process from all three perspectives leads to better results. More analysis in Appendix D.3.

We further conducted hyper-parameter experiments to validate the choice of the number of blocks $N_2$ and the attention heads $M_2$ in the graph transformer, and selected sets of configurations as $con1$, $con2$ ..., were compared with baseline methods in terms of training time and encoding time. Training time is measured over 100 epochs (hours), and encoding time is measured for a single pass over the query set (ms). These experiments are shown in Fig. 6. To combine performance and efficiency, we chose $con1$ as the final parameter, i.e., $N_2 = 2, M_2 = 4$.

### 4.6 MODULE GENERALIZATION

Our proposed method serves as an information-rich strategy that provides broad support for cross-modal training. To validate this, we extend it to the discriminative approach DHaPH and DDBH. As shown in Fig. 7, our method can be used in a plug-and-play manner to support various approaches. Furthermore, to validate the capacity of augmentation-based methods to cope with low-information training in challenging environments, We first halve the train set size and then halve the batch size consecutively. Considering the instability in this scenario, we perform multiple experiments and record the variance as shown in Fig. 8. Our method can still stably provide discriminative information to support training in the face of fewer samples. We visualize the distribution of negative samples before and after the proposed method generates difficult negative samples in Fig. 9. We also checked the stability of the backbone in Appendix D.4.

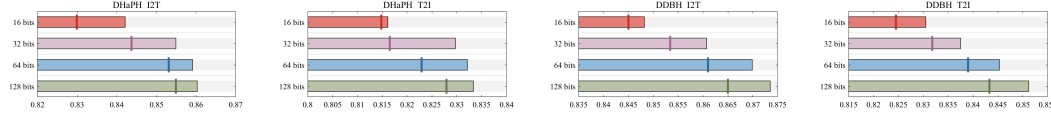

Figure 7: Bullet chart visualization on MIRFLICKR-25K. The target markers indicate the baseline and bars correspond to add the DGHDGH module.

### 4.7 NOISE ROBUSTNESS

We conducted noise label experiments to verify the performance degradation of the model when countering label interference. Noise rates of 0.2, 0.5, and 0.8 indicate random two-digit label inversion of the corresponding proportion of samples during training, follow in Wang et al. (2024).

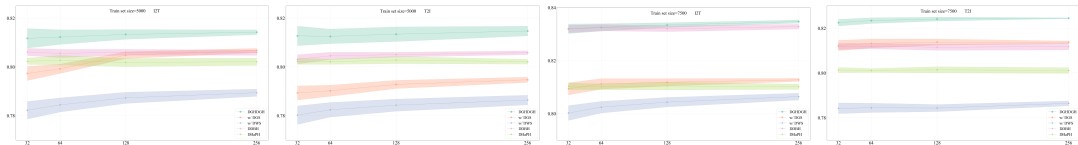

Figure 8: Batch stability error with line plots for different training set sizes (5000, 7500). Batch size is taken as 32, 64, 128, 256.

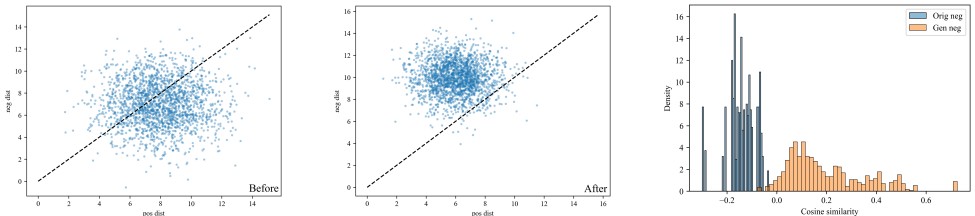

Figure 9: Visualization of the distribution of relative distances of negative samples before and after generation, and their cosine similarity histograms.

DGHDGH maintains stable retrieval performance at all noise levels and consistently outperforms all baselines. Graph propagation aggregates the relationship signals of multiple adjacent samples and dominates the propagation process with global relationships, forming a natural denoising filter that suppresses the influence of damaged labels. Meanwhile difficulty-aware synthetic negative samples generate hard negative samples based on cross-modal similarity and structural consistency, without relying on original labels, thus avoiding the amplification of noise and the generation of misleading negative samples by erroneous labels. Because synthetic negatives does not depend on noisy labels but on learned similarity and global propagation, it is inherently more robust than baselines.

Table 4: mAP results (%) of DDGRH and baselines under different noise rates on the MIRFLICKR-25K dataset w.r.t. four bits.

| Task | Method | 16 bits | | | 32 bits | | | 64 bits | | | 128 bits | | |
|---|---|---|---|---|---|---|---|---|---|---|---|---|---|
| | | *0.2* | *0.5* | *0.8* | *0.2* | *0.5* | *0.8* | *0.2* | *0.5* | *0.8* | *0.2* | *0.5* | *0.8* |
| Image ↓ Text | NRCH | 76.38 | 75.61 | 73.24 | 76.75 | 76.21 | 74.76 | 77.05 | 76.98 | 76.07 | 77.83 | 77.10 | 76.11 |
| | DNPH | 77.70 | 75.55 | 75.37 | 79.74 | 80.16 | 79.04 | 81.67 | 82.44 | 81.01 | 83.53 | 83.32 | 81.08 |
| | DHaPH | 83.73 | 81.64 | 78.04 | 82.94 | 78.11 | 75.82 | 82.10 | 79.17 | 76.85 | 82.64 | 79.77 | 77.28 |
| | BiLGSEH | 78.11 | 77.35 | 75.93 | 78.24 | 77.36 | 75.15 | 80.72 | 78.42 | 76.30 | 80.00 | 79.32 | 77.46 |
| | DPBE | 77.85 | 77.16 | 73.79 | 80.8 | 77.16 | 74.09 | 82.21 | 78.28 | 75.45 | 84.12 | 80.11 | 76.77 |
| | DDBH | **83.79** | **82.20** | 76.50 | **84.70** | 82.11 | 78.78 | 85.04 | 84.58 | 79.46 | 85.64 | 84.61 | 80.29 |
| | **DGHDGH** | 81.95 | 81.78 | **78.65** | 84.06 | **82.76** | **80.89** | **85.96** | **84.71** | **82.64** | **85.96** | **85.15** | **84.16** |
| Text ↓ Image | NRCH | 74.55 | 74.31 | 72.20 | 75.53 | 74.68 | 72.59 | 75.88 | 75.41 | 74.59 | 75.71 | 75.80 | 74.64 |
| | DNPH | 76.40 | 75.25 | 75.18 | 78.56 | 79.28 | 77.05 | 80.56 | 80.18 | 79.25 | 81.26 | 80.76 | 79.45 |
| | DHaPH | 81.51 | 79.49 | 77.66 | 80.52 | 75.96 | 73.22 | 78.69 | 76.39 | 74.23 | 79.31 | 76.99 | 74.38 |
| | BiLGSEH | 77.23 | 76.61 | 76.30 | 81.27 | 77.85 | 72.16 | 79.21 | 75.67 | 75.14 | 79.55 | 77.36 | 76.72 |
| | DPBE | 76.26 | 75.82 | 73.29 | 78.83 | 76.28 | 74.38 | 81.17 | 78.13 | 75.76 | 82.59 | 80.14 | 77.22 |
| | DDBH | **82.16** | 80.27 | 77.07 | 82.34 | 81.22 | 79.62 | 84.24 | 82.79 | 79.77 | 83.76 | 82.69 | 79.84 |
| | **DGHDGH** | 81.03 | **80.42** | **78.80** | **82.59** | **81.67** | **80.87** | **84.56** | **83.50** | **82.09** | **84.56** | **84.13** | **83.35** |

The best and second-best performances are highlighted in boldface and underlined.

## 5 CONCLUSION

In this work we presented DGHDGH, the first framework that introduces discriminative generation into deep hashing. By combining global relational modeling with hardness-adaptive synthesis, our method generates semantically consistent negatives that sharpen decision boundaries in Hamming space. Extensive experiments on multiple benchmarks verify that DGHDGH significantly improves retrieval accuracy and discriminability over state-of-the-art baselines. Beyond its standalone performance, our framework is modular and can serve as a plug-and-play enhancement for existing cross-modal approaches,and will be accessed for arbitrary representation learning in the future.

## 6 ACKNOWLEDGEMENT

This work was supported by the National Natural Science Foundation of China (No.62302338), Shandong Province Natural Science Foundation (No.ZR2022QF046, No.ZR2025MS1067), Natural Science Foundation of Chongqing (No.CSTB2023NSCQ-MSX0407), and Science and Technology Research Program of Chongqing Municipal Education Commission (No.KJQN202500511).

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

# A  RELATED WORK

This section provides the detailed discussion of related works that were briefly summarized in Section 2 of the main paper.

## A.1  DEEP CROSS-MODAL HASHING RETRIEVAL

Deep Cross-modal Hashing Retrieval (DCHR) aims to map heterogeneous modalities,such as images and text, into a collaborative Hamming space end-to-end for efficient approximate nearest neighbor retrieval Chen et al. (2023). With the benefit of low storage cost and high retrieval efficiency, DCHR methods have attracted wide interest in the field of cross-modal retrieval, achieving superior similarity retrieval performance Li et al. (2023); Zhu et al. (2023). Deep Cross-Modal Hashing (DCMH) uses negative log-likelihood loss to maximize the similarity of hash codes for similar samples and minimize it for dissimilar samples Jiang & Li (2017). Self-Supervised Adversarial Hashing (SSAH) uses adversarial learning and self-supervised semantic discovery to improve the alignment of multi-label semantic distributions Li et al. (2018a). Multimodal transformers with

a differentiable hashing mechanism are leveraged by Differentiable Cross-modal Hashing via Multimodal Transformers (DCHMT), enabling gradient-based optimization through CLIP-style representations Tu et al. (2022).

In recent years, researchers have sought to overcome the limitations of conventional cross-modal hashing by incorporating more informative learning strategies. Deep Neighborhood-preserving Hashing (DNpH) improves the discrimination and semantic consistency of cross-modal hashing by preserving the Neighborhood structure and combining quadratic spherical mutual information Qin et al. (2024). By introducing hierarchical agent and self-paced learning mechanism, Deep Hierarchy-aware Proxy Hashing (DHaPH) can gradually capture global and local hierarchical semantic information Huo et al. (2024b). Deep Evidential Hashing for Trustworthy Cross-Modal Retrieval (DECH) models the uncertainty information by evidence theory, which makes cross-modal hashing more advantageous in generating trustworthy and interpretable retrieval results Li et al. (2025c).

## A.2 INFORMATIVE LEARNING

A core bottleneck for cross-modal hashing is that training is easily dominated by uninformative easy samples, causing slow convergence and weak decision boundaries in Hamming space Qin et al. (2022); Meng et al. (2021). Informative learning tackles this by prioritizing supervision that carries higher training value, either by mining difficult instances from existing data or by generating challenging instances to enrich supervision. From this perspective, The past works can be summarized as the pursuit of more informative learning and divided into two major threads, mining-based and augmentation-based.

### A.2.1 MINING-BASED LEARNING

At the earliest, facenet recognized the importance of mining and proposed semi-hard sampling to select informative examples for the triplet loss Schroff et al. (2015). Subsequently, various sampling modalities oriented to specific regions have blossomed. Easy positive mining approach holds that the query sample only needs to be close to the simple examples among its positive samples rather then whole positives Xuan et al. (2020b). This relaxed information filtering mechanism leads to better generalization. Hard negative mining is a major focus of mining, which aims to select difficult negatives that are highly similar to positives Bucher et al. (2016); Simo-Serra et al. (2015); Suh et al. (2019); Xuan et al. (2020a). Learning this information in a targeted manner during training can enhance the model's ability to distinguish between positive and negative examples. Hard example mining goes one step further by selecting both difficult negative examples and positive examples (i.e., positive examples with low similarity). A high degree of information acquisition allows the model to draw a clear line between positives and negatives Rao et al. (2024); Shrivastava et al. (2016); Smirnov et al. (2018). Distance Weighted Sampling (DWS) point out that the mining method is limited by the narrow area selected, and the reduction of the selected sample size affects the amount of information obtained. Extensive sampling can not only lead to an improvement in the amount of information, but also achieve better generalization by learning different distance relationships, brings the same or even higher performance impact as the loss function Wu et al. (2017). Back to the cross-modal domain, Triplet-based Deep Hashing (TDH) introduces the Triplet with hard mining into hashing, so as to improve the discrimination of cross-modal similarity ranking Deng et al. (2018). Hard-Negative Selection Strategy (HNSS) using the improved marginal ranking loss to examined a new strategy for hard-negative mining in cross-modal retrieval Galanopoulos & Mezaris (2021). This also pointed out that mining methods are limited by batch size even train dataset size and lack enough imformation. This leads to overfitting or sub-optimization in the end. In this scenario, a series of methods designed to provide additional information have emerged.

### A.2.2 AUGMENTATION-BASED LEARNING

**Generator-based** The most common generation method is the Generative Adversarial Network (GAN). After its popularity, many methods have also attempted to utilize GAN to enhance hash learning Qian et al. (2023). While emphasizing self-supervision, Self-Supervised Adversarial Hashing (SSAH) uses adversarial network mechanisms to enforce cross-modal consistency Li et al. (2018a). HashGAN introduce the generative attention mask and adversarial sample generation,

improves the discriminative ability of hash representation by generating a network to interfere with the discriminator Cao et al. (2018b).

**Memory-based** A range of methods use memoization module like memory bank or backup queue to get around the batch size limit to get more information. The embeddings of previous iterations are maintained by Cross Batch Memory and considered to be still informative in the current batch Wang et al. (2020b). Fast Partial-Modal Online Cross-Modal Hashing (FPO-CMH) facilitates efficient online cross-modal hash learning by using a multimodal dual-tier anchor bank Li et al. (2025a).

**Interpolation-based** Mixup proposes a linear interpolation method of input and label to generate virtual samples Zhang et al. (2017). This approach has been widely used to improve generalization and adversarial robustness. DAS reiterates the "miss embedding" problem for the mining and interpolates all real embeddings as anchors to generate positive and negative pseudo-embeddings Liu et al. (2022).

**Hard Negative Generation** To targeted acquisition negative samples with a larger amount of information, the hard negative generation (HNG) can be regarded as a special direction. Deep adversarial metric learning (DAML) synthesizes simple negatives into hard negatives through adversarial trainingDuan et al. (2018). Hardness-aware Deep Metric Learning (HDML) performs hardness-aware interpolation between anchor-negative pairs and then uses an autoencoder to generate corresponding features Zheng et al. (2019). A two-stage synthesis framework is introduced to generate hard positives and hard negatives at the same time Zhao et al. (2018). Most of these methods obtain sample relationships through interpolation and features through generator. However, due to the inherent shortcomings of interpolation methods, difficult sample generation is rarely applied to cross-modal hashing, as the synthesized difficult negative samples also lack spatial feature perception, which is useless or even interferes with cross-modal semantic alignment.

Motivated by these limitations, we propose a novel method for generating difficult samples that can be applied to cross-modal hashing, which can assist in cross-modal semantic alignment by obtain spatial feature perception.

## B  THEORETICAL INTERPRETATION OF RGP PROPAGATION

The behavior of RGP propagation can be understood by comparing its update rule with the classical graph smoothing models. Standard GNN propagation is often approximated by a Laplacian operator of the form:

$$H^{(l+1)} = \alpha H^{(l)} + (1 - \alpha) D^{-1/2} A D^{-1/2} H^{(l)} \tag{17}$$

which has been formally shown to act as a low-pass filter on graph signals and to cause feature convergence across connected nodes as depth increases. This phenomenon has been documented in several analyses of message-passing networks, including studies of over-smoothing where node embeddings tend to collapse into indistinguishable representations when eigenvalues of the propagation operator suppress high-frequency components. Representative discussions include the analyses of Oono & Suzuki (2019) and Li et al. (2018b), which indicate that repeated Laplacian-style propagation inevitably produces smoothing behavior.

The update rule of RGP differs from this model. Let $A_p$ denote the semantic positive graph and $A_n$ denote the synthesized negative graph constructed during training. RGP applies masked attention in which each anchor attends only to nodes identified by $A_n$, while entries in $A_p$ are suppressed. The propagation can be written as

$$H^{(l+1)} = H^{(l)} + \eta \operatorname{softmax}(M_n H^{(l)} W) H^{(l)} \tag{18}$$

where $M_n$ is a masking operator that selects negative nodes and $\eta$ is a small scale factor. Because $M_n$ excludes positive edges entirely, the operator does not approximate a Laplacian. The update term contains no averaging over semantic neighbors. Instead, it introduces a structured repulsive transformation that pushes representations away along directions defined by synthesized hard negatives. This behavior is closer to contrastive propagation than to classical smoothing. Unlike Laplacian filters that attenuate high-frequency components, the negative-aware update amplifies structural

distinctions in the feature space because the gradient of the attention logits depends on differences between the anchor and its negative set.

A direct consequence is that the eigenstructure governing the propagation is not dominated by the low-frequency bases of the graph Laplacian. The update matrix produced by masked attention is asymmetric and does not share the spectrum of $D^{-1/2}AD^{-1/2}$. Prior analyses of attention-based propagation observe that attention operators do not behave as low-pass filters Verma & Zhang (2019). In the case of RGP, the combination of masked attention and synthesized negatives produces a directional transformation that retains discriminative structure even after multiple layers. This theoretical form explains why RGP avoids the collapse and homogeneity commonly associated with Laplacian-based architectures.

## C  EXPERIMENTS SETTINGS

### C.1  IMPLEMENTATION DETAILS

Based on RTX A6000 Ada GPUs, we adopt the open-source PyTorch framework to implement our proposed DGHDGH algorithm and other methods Paszke et al. (2019). The PyTorch version is 2.3.0. They are all performed in a unified experimental setting. The pre-training parameters of the Transformer encoders in feature extraction are reference in CLIP ViT-B/32 from Radford et al. (2021) applied on all methods, which have 12 Transformer blocks, and 8 heads for each attention module in blocks. For baselines, we follow their official implementations where available, or adopt hyper-parameters from the original papers.

For parameter optimization, We use Adam as the optimizer, where the learning rate are 1e-4 for feature extraction with hash layers, and 1e-5 for graph transformer. The weight decay is 0.2 and the batch size is set to 128. Besides, for image preprocessing, we resize to $224 \times 224$, center crop, and normalize with CLIP's default statistics. Texts are tokenized using the CLIP tokenizer with a maximum length of 77. We evaluate hash codes of 16, 32, 64, and 128 bits.

### C.2  EVALUATION METRICS

In this work, we use mean Average Precision (mAP) that comprehensive retrieval evaluation, and fisher ratio and Precision within Hamming Radius $\leq 2$ ($P@H \leq 2$) to evaluate the discrimination of models. Moreover, we introduce Normalized Discounted Cumulative Gain (NDCG), Precision–Recall Curve (PR), and Top-K Precision Curve (P@K) in the appendix to further validate the retrieval performance of the model.

#### C.2.1  MEAN AVERAGE PRECISION

We primarily use mAP as a performance metric, which calculates the average precision of each sample in the query set retrieved from the database set and then averages it again. The mAP is the average precision under different recall thresholds, and it is a comprehensive retrieval evaluation including recall and precision.The formula is shown as:

$$mAP@K = \frac{1}{n}\sum_{i=1}^{n} AP_i@K, \quad \text{where } AP_i@K = \frac{1}{K}\sum_{j=1}^{k} \frac{r_j}{j} \times l_{ij}. \tag{19}$$

When $i, j$ belong to the same category, $l_{ij} = 1$, otherwise $l_{ij} = 0$. $r_j$ represents the number of relevant samples in top-$j$ in the ranking list. $n$ is the number of query samples. In this paper, we choose $k = all$ i.e. the number of database samples. Among them, mAP I2T uses image modality for query and text modality for database, T2I is similar.

#### C.2.2  PRECISION WITHIN HAMMING RADIUS $\leq 2$

To directly measure retrieval quality in Hamming space, we also compute Precision within Hamming radius $\leq 2$ ($P@H \leq 2$) Liu et al. (2012). For a given query, this metric counts the proportion of

relevant items among all retrieved samples whose Hamming distance to the query is less than or equal to 2. As shown in formula:

$$P@H \leq 2 = \frac{\text{Number of relevant retrieved items within } H \leq 2}{\text{Total Number of retrieved items within } H \leq 2} \tag{20}$$

This metric reflects the local discriminative capability of hash codes in a compact neighborhood. A higher $P@H \leq 2$ means that the retrieved neighbors are more semantically consistent with the query. This reflect how effectively the learned hash codes preserve semantic neighborhood structures, and a higher PH2 indicates stronger local discrimination within the Hamming space.

### C.2.3 FISHER RATIO

Finally, to quantitatively assess the discriminability of hash codes, we compute the Fisher ratio, which compares the separability of positive and negative pairs in Hamming space. Specifically:

$$Fisher = \frac{\mu_{neg} - \mu_{pos}}{\sigma'}, \quad \text{where } \sigma' = \sqrt{\frac{\sigma_{pos}^2 + \sigma_{neg}^2}{2}}. \tag{21}$$

where $\mu_{neg}$ and $\mu_{pos}$ denote the mean Hamming distances of negative and positive pairs, and $\sigma'$ is the pooled standard deviation by the std $\sigma_{neg}$ and $\sigma_{pos}$.

In practice, we compute Fisher ratios by randomly sampling 50, 100, 200, and 400 pairs of positive and negative examples, from the hash codes of database set after training while anchors are from query set. We repeating each sampling 5 times, and reporting the averaged results with standard deviations. This statistic quantifies the margin between positive and negative relations, and a larger Fisher ratio demonstrates improved global separability and stronger discriminative structure in the Hamming space.

### C.2.4 NORMALIZED DISCOUNTED CUMULATIVE GAIN (NDCG)

To further evaluate ranking quality in cross-modal retrieval, we report the Normalized Discounted Cumulative Gain (NDCG), which measures how well a retrieval method ranks relevant items near the top of the list. For a given query, the Discounted Cumulative Gain (DCG) at rank $K$ is defined as

$$DCG@K = \sum_{j=1}^{K} \frac{rel_j}{\log_2(j+1)}, \tag{22}$$

where $rel_j$ is the relevance score of the item at rank $j$. The ideal DCG (IDCG) is obtained by sorting relevant items in the optimal order. The normalized form is

$$NDCG@K = \frac{DCG@K}{IDCG@K}. \tag{23}$$

NDCG emphasizes the importance of ranking correct items earlier in the retrieval list. A higher NDCG indicates that the model places semantically relevant samples closer to the top positions, reflecting high-quality retrieval ordering beyond binary relevance. In this work, we choice $k = 1000$ i.e. NDCG@1000.

### C.2.5 PRECISION–RECALL CURVE

We also use the Precision–Recall (PR) curve to visualize the trade-off between precision and recall across different similarity thresholds in the Hamming space. For each threshold, the retrieved set is determined by selecting items whose Hamming distance to the query is below the threshold. Precision and recall are computed as

$$Precision = \frac{TP}{TP + FP}, \qquad Recall = \frac{TP}{TP + FN}, \qquad (24)$$

where $TP$, $FP$, and $FN$ are the number of true positives, false positives, and false negatives. PR curves provide a continuous view of retrieval behavior under varying cutoff distances, allowing us to assess how a model handles both strict and loose retrieval requirements. A curve that stays consistently high indicates stable performance across a wide range of thresholds.

### C.2.6 TOP-$K$ PRECISION CURVE

To measure performance at different retrieval depths, we compute the Top-$K$ Precision curve. For each query, the precision at rank $K$ is defined as

$$P@K = \frac{1}{K} \sum_{j=1}^{K} l_{ij}, \qquad (25)$$

where $l_{ij} = 1$ if the $j$-th retrieved item is relevant and $0$ otherwise. By sweeping $K$ from small to large values, the Top-$K$ precision curve shows how the method behaves when retrieving only a few nearest neighbors or when retrieving deeper lists. This metric reflects the stability of the semantic preservation across different retrieval lengths. Higher curves imply that the learned hash codes remain discriminative regardless of the retrieval depth.

## D ADDITIONAL EXPERIMENTS

### D.1 COMPARISON WITH CLASSICAL METHODS ON THE STANDARDIZED BACKBONE

To further validate the effectiveness of our approach, we adapted the CLIP ViT-B/32 as the standardized feature extractor for classical methods published prior to CLIP, which did not utilize Transformer architectures in their network, namely Deep Cross-Modal Hashing (DCMH) Jiang & Li (2017), Cross-Modal Hamming Hashing (CMHH) Cao et al. (2018a), Adversary Guided Asymmetric Hashing (AGAH) Gu et al. (2019), Deep Adversarial Discrete Hashing (DADH) Bai et al. (2020), Self-constraining attention hashing network (SCAHN) Wang et al. (2020a), and add '-T' to indicate backbone replacement. As shown in Tab. 5, Our method continues to deliver superior performance under this standardized evaluation, demonstrating that its effectiveness is not tied to specific feature extraction techniques but rather stems from the overall learning framework and its design innovations.

### D.2 COMPARISON WITH BASELINE ON MORE METRICS

To comprehensively evaluate retrieval performance, we also present comparisons of NDCG, PR curves, and P@K curves, as shown in Tab. 6, Fig. 10, and Fig. 11 respectively. Experiments show that, except for some cases, DGHDGH still maintains a broad lead in these indicators. This indicates that our method clearly has better retrieval performance.

### D.3 HYPER-PARAMETER ANALYSIS

After the Ablation study of three loss coefficient terms $\gamma_{is}$, $\gamma_{sp}$ and $\gamma_{cd}$, we further explore the influence of different values of them on retrieval performance, as shown in Fig. 12. As $\gamma_{is}$ a similarity constraint, the gradient is naturally the largest, so the choice of the anchor scale is a natural choice to ensure the stability of the optimization. We set $\gamma_{is}$ to 1 and adjust the other two items from 0.1 to 10. We applied the parameter configuration at the best performance to the experiments while $\gamma_{sp} = 1$ and $\gamma_{cd} = 0.2$.

Table 5: mAP@all results(%) of DGHDGH with classic methods on the three benchmark datasets w.r.t. four hash bits.

| Task | Reference | Method | MIRFLICKR-25K | | | | NUS-WIDE | | | | MS COCO | | | |
|---|---|---|---|---|---|---|---|---|---|---|---|---|---|---|
| | | | 16 | 32 | 64 | 128 | 16 | 32 | 64 | 128 | 16 | 32 | 64 | 128 |
| Image ↓ Text | CVPR'17 | DCMH | 76.87 | 77.36 | 77.97 | 78.81 | 53.79 | 55.13 | 56.17 | 57.03 | 53.99 | 54.44 | 56.27 | 56.59 |
| | | *DCMH-T* | 82.78 | 84.23 | 84.01 | 85.17 | 63.97 | 66.78 | 68.59 | 68.64 | 62.74 | 64.99 | 67.20 | 67.59 |
| | ECCV'18 | CMHH | 69.32 | 69.79 | 69.84 | 70.23 | 54.39 | 55.46 | 55.2 | 52.91 | 51.45 | 45.09 | 52.09 | 50.21 |
| | | *CMHH-T* | 81.11 | 82.04 | 82.52 | 83.08 | 67.41 | 68.88 | 69.28 | 69.71 | 54.90 | 59.46 | 64.30 | 65.49 |
| | ICMR'19 | AGAH | 72.48 | 72.17 | 71.95 | 72.82 | 39.45 | 41.07 | 42.58 | 43.58 | 55.01 | 55.15 | 55.18 | 55.54 |
| | | *AGAH-T* | 80.79 | 81.22 | 81.98 | 83.49 | 59.81 | 64.69 | 66.44 | 67.56 | 61.45 | 64.16 | 65.94 | 66.55 |
| | ICMR'20 | DADH | 80.98 | 81.62 | 81.93 | 82.17 | 63.5 | 65.68 | 65.46 | 66.61 | 59.52 | 61.18 | 62.37 | 63.24 |
| | | *DADH-T* | 82.30 | 83.23 | 84.58 | 84.03 | 66.54 | 67.98 | 68.71 | 70.58 | 63.39 | 66.51 | 68.76 | 69.27 |
| | Neuco'20 | SCAHN | 79.55 | 82.48 | 82.97 | 82.88 | 64.63 | 66.16 | 66.45 | 65.49 | 63.76 | 64.75 | 65.19 | 61.49 |
| | | *SCAHN-T* | 82.13 | 84.32 | 84.87 | 85.31 | 68.29 | 68.01 | 69.13 | 70.23 | 67.72 | 68.09 | 70.12 | 69.88 |
| | **Ours** | **DGHDGH** | **84.66** | **86.19** | **87.13** | **87.75** | **69.72** | **71.68** | **72.60** | **73.76** | **72.06** | **74.71** | **77.13** | **79.19** |
| Text ↓ Image | CVPR'17 | DCMH | 78.57 | 79.98 | 80.29 | 80.83 | 57.47 | 58.1 | 58.53 | 59.04 | 52.71 | 54.24 | 54.5 | 55.26 |
| | | *DCMH-T* | 80.88 | 82.01 | 83.11 | 84.09 | 64.24 | 66.33 | 69.17 | 69.74 | 62.98 | 65.53 | 67.00 | 67.92 |
| | ECCV'18 | CMHH | 71.81 | 71.04 | 72.94 | 73.72 | 49.56 | 48.31 | 48.2 | 47.81 | 49.1 | 49.3 | 48.89 | 50.15 |
| | | *CMHH-T* | 78.33 | 79.51 | 80.16 | 81.75 | 68.53 | 69.67 | 70.04 | 70.46 | 55.33 | 58.17 | 64.81 | 65.52 |
| | ICMR'19 | AGAH | 70.82 | 71.82 | 73.44 | 74.38 | 43.44 | 39.8 | 43.82 | 44.05 | 50.12 | 51.46 | 51.91 | 51.36 |
| | | *AGAH-T* | 79.52 | 80.10 | 82.12 | 82.61 | 60.10 | 64.38 | 67.43 | 68.17 | 60.91 | 64.38 | 64.49 | 65.91 |
| | ICMR'20 | DADH | 80.19 | 81.01 | 81.37 | 81.35 | 61.11 | 61.82 | 62.18 | 63.26 | 56.49 | 57.9 | 58.7 | 60.37 |
| | | *DADH-T* | 81.42 | 82.32 | 83.20 | 82.65 | 68.06 | 68.82 | 69.18 | 70.54 | 61.88 | 66.50 | 68.37 | 68.84 |
| | Neuco'20 | SCAHN | 78.26 | 80.66 | 80.64 | 80.66 | 65.87 | 66.26 | 66.48 | 66.09 | 63.77 | 65.12 | 64.93 | 61.45 |
| | | *SCAHN-T* | 81.26 | 82.38 | 82.55 | 83.06 | 68.97 | 69.14 | 70.18 | 70.49 | 67.36 | 68.52 | 69.57 | 70.05 |
| | **Ours** | **DGHDGH** | **83.03** | **84.21** | **85.09** | **85.74** | **70.75** | **72.64** | **73.75** | **74.64** | **71.16** | **74.69** | **77.41** | **79.59** |

The best and second-best performances are highlighted in boldface and underlined. *Neuco* denotes *Neurocomputing*.

Table 6: NDCG@1000 results(%) of DGHDGH with baselines on three benchmark datasets w.r.t. four hash bits.

| Task | Method | MIRFLICKR-25K | | | | NUS-WIDE | | | | MS COCO | | | |
|---|---|---|---|---|---|---|---|---|---|---|---|---|---|
| | | 16 | 32 | 64 | 128 | 16 | 32 | 64 | 128 | 16 | 32 | 64 | 128 |
| Image ↓ Text | DNpH | 50.22 | 52.66 | 54.30 | 55.07 | 50.86 | 53.37 | 55.26 | 55.78 | 33.88 | 40.06 | 41.07 | 42.19 |
| | DHaPH | 46.29 | 48.62 | 48.16 | 48.29 | 51.23 | 46.24 | 54.03 | 55.20 | 39.19 | 41.37 | 41.40 | 42.42 |
| | BiLGSEH | 47.71 | 50.21 | 51.13 | 51.05 | 54.07 | 54.75 | 56.92 | 57.66 | 46.50 | 50.22 | 50.83 | 50.43 |
| | DECH | 40.89 | 49.34 | 49.46 | 49.29 | 46.40 | 54.05 | 53.23 | 56.32 | 19.89 | 24.92 | 30.58 | 32.41 |
| | DPBE | 41.56 | 46.72 | 49.67 | 50.99 | 44.29 | 45.25 | 47.66 | 47.08 | 41.96 | 44.62 | 49.81 | 52.10 |
| | DDBH | 51.25 | 52.72 | 52.42 | 52.57 | 52.88 | 54.89 | 56.56 | 56.09 | 38.48 | 42.56 | 46.04 | 47.78 |
| | **DGHDGH** | **53.26** | **55.19** | **56.11** | **57.38** | 51.68 | **55.95** | **58.59** | **60.94** | 37.31 | 42.52 | **51.10** | **54.72** |
| Text ↓ Image | DNpH | 42.30 | 45.17 | 46.15 | 47.06 | 49.74 | 50.67 | 52.87 | 52.77 | 33.32 | 40.04 | 41.26 | 42.81 |
| | DHaPH | 45.07 | 41.32 | 42.12 | 43.45 | 46.63 | 44.16 | 48.42 | 49.15 | 40.49 | 40.86 | 43.62 | 43.32 |
| | BiLGSEH | 45.53 | 47.09 | 48.18 | 48.60 | 53.24 | 54.06 | 55.07 | 55.93 | 46.43 | 50.46 | 51.59 | 51.06 |
| | DECH | 38.69 | 41.53 | 42.86 | 42.84 | 47.58 | 54.82 | 53.23 | 53.46 | 20.52 | 25.86 | 30.62 | 32.10 |
| | DPBE | 34.16 | 39.05 | 41.52 | 42.46 | 41.06 | 43.59 | 44.88 | 44.60 | 33.83 | 47.20 | 49.89 | 52.45 |
| | DDBH | 42.63 | 43.50 | 44.10 | 44.56 | 50.98 | 51.52 | 52.93 | 54.46 | 38.79 | 40.99 | 44.56 | 46.48 |
| | **DGHDGH** | **46.40** | **47.52** | **49.46** | **51.16** | 51.42 | 53.35 | **56.11** | **57.12** | 36.71 | 46.59 | **52.90** | **56.14** |

The best and second-best performances are highlighted in boldface and underlined.

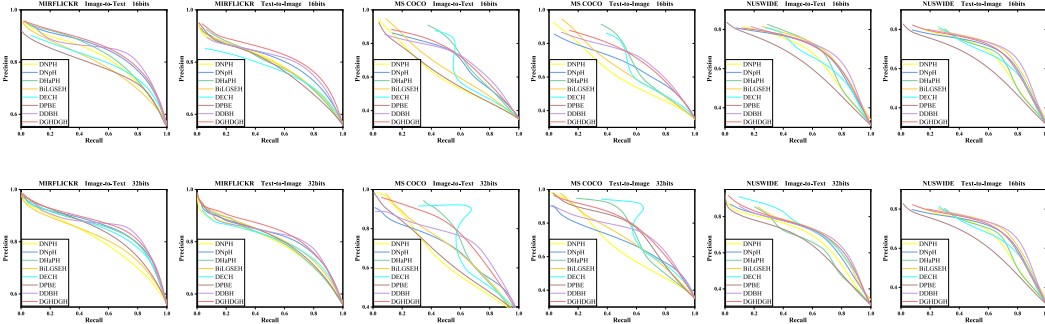

Figure 10: Results of Precision-Recall curves on three benchmark datasets w.r.t. 16 and 32 bits.

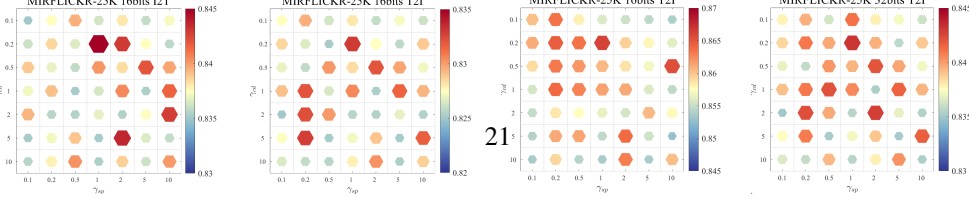

Figure 12: Hyper-parameter analysis on MIRFLICKR-25K w.r.t. 16 and 32 bits.

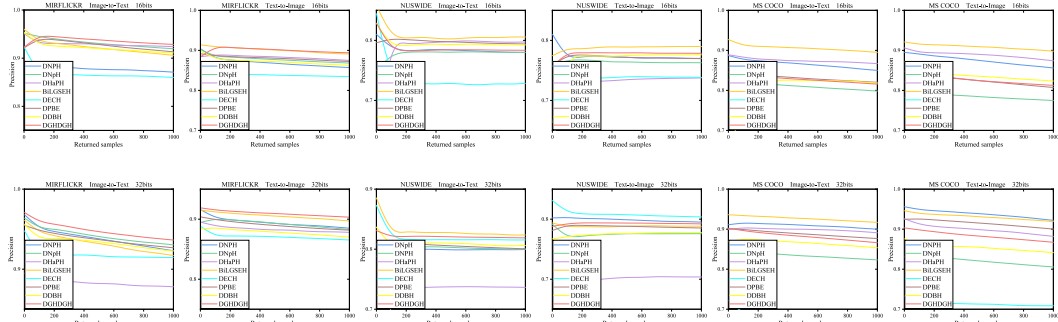

Figure 11: Results of Top-K Precision curves on three benchmark datasets w.r.t. 16 and 32 bits.

## D.4 BACKBONE GENERALIZATION

To judge the generalization across different backbones, we further investigate several CLIP architecture with its variant BLIP and SigLIP2 Li et al. (2022); Tschannen et al. (2025). As summarized in Tab. 7, we report the **I2T** and **T2I** mAP (%) on MIFLICLR-25K 16 bits, along with other metrics: the number of **Parameters** in each backbone (in *Million*), total **Training Time** for 100 epochs training (in *Hour*), **Encoding Time** for the query dataset encoding (in *Millisecond*), and the **Converge Epoch** indicating the best performance within 100 epochs. These results collectively demonstrate the strong generalization capability of our proposed DGHDGH method across diverse backbone architectures.

Table 7: Various metrics of DGHDGH with different backbones on the MIRFLICKR-25K dataset w.r.t. 16 bits.

| Metric | CLIP | | | BLIP | | SigLIP2 | | |
|---|---|---|---|---|---|---|---|---|
| | *ViT-B/32* | *ViT-L/14* | *Res50/16* | *I-C-Base* | *I-C-Large* | *Base-16/224* | *Base-32/256* | *Large-16/256* |
| **Parameters** | 151.3 | 427.6 | 291.0 | 224.7 | 447.2 | 375.2 | 376.9 | 881.5 |
| **Image→Text** | 84.66 | 84.82 | 83.22 | 82.96 | 83.83 | 82.54 | 84.82 | 84.50 |
| **Text→Image** | 83.03 | 83.25 | 82.62 | 83.01 | 81.20 | 80.95 | 83.25 | 84.04 |
| **Training Time** | 1.206 | 4.652 | 5.312 | 5.945 | 13.28 | 1.753 | 2.018 | 5.374 |
| **Encoding Time** | 27.05 | 37.98 | 32.41 | 72.57 | 126.8 | 34.46 | 31.05 | 47.39 |
| **Converge Epoch** | 44 | 56 | 61 | 78 | 69 | 83 | 72 | 75 |

