# OpenReview forum: "Deep Global-sense Hard-negative Discriminative Generation Hashing for Cross-modal Retrieval"
_ICLR.cc/2026/Conference — ICLR 2026 Poster_

### Official Review · Reviewer_Xgam · 2025-10-20

**Soundness:** 3
**Presentation:** 3
**Contribution:** 3
**Rating:** 6
**Confidence:** 4

**Summary:**

This paper addresses the challenge of discriminative cross-modal hashing through a global-sense perspective, constructing adaptive hard negatives that reflect global semantics rather than local feature proximity. The approach builds upon two main components: a graph propagation network (RGP) for capturing higher-order semantic dependencies and a discriminative synthesis unit (DGS) that regulates interpolation difficulty via channel-wise weighting. Conceptually, this work reframes hard-negative generation as an optimization over a semantic manifold.

**Strengths:**

1. Introduces a coherent, theoretically inspired motivation for rethinking negative sampling as a global consistency problem.
2. The loss design reflects an interesting interplay between semantic preservation, interpolation similarity, and coefficient diversity.
3. Empirical results validate the conceptual claims with strong mAP improvements.

**Weaknesses:**

1. In Figure 3, the comparison between DGHDGH and DHaPH shows large performance gains, but it is unclear whether both models use identical backbones and training setups. A controlled experiment would be necessary to ensure fairness.
2. The RGP module remains largely intuitive, while its empirical benefits are evident, there is no theoretical analysis of how the propagation maintains information fidelity or prevents over-smoothing.
3. The DGS module’s channel-wise λ weights perform well, but their dynamics are not visualized. A λ-distribution plot or feature-space interpolation visualization would clarify how “difficulty” is being modulated.
4. The method is validated only for image-text retrieval. Can DDGSH extend to audio/video modalities?

**Questions:**

1. Could the authors mathematically relate the RGP operation to spectral diffusion or Laplacian smoothing?
2. How sensitive is the overall model to λ initialization?
3. Discussing the future work like audio/video retrieval in Conclusion section would strengthen the impact.

---

> ### Author Response · Authors · 2025-11-21
> **Thanks for your review**
>
> Dear Reviewer Xgam,
>
> Thank you for your careful assessment and for raising several valuable questions regarding evaluation fairness, the theoretical behavior of RGP, the interpretability of the learned coefficients, and the broader applicability of the proposed framework. Your remarks improved both the technical clarity and the broader framing of our work. We respond to each of your concerns in detail in the following sections.

---

> ### Author Response · Authors · 2025-11-21
>
> > W1: In Figure 3, the comparison between DGHDGH and DHaPH shows large performance gains, but it is unclear whether both models use identical backbones and training setups. A controlled experiment would be necessary to ensure fairness.
>
> A1: Regarding the fairness of the comparison in Figure 3, all competing methods, including DGHDGH and DHaPH, are trained under an identical experimental configuration. Specifically, every method uses the same CLIP-ViT-B/32 backbone, the same hash length, the same optimizer, and the same batch size and learning schedule on each dataset. The comparison isolates only the contribution of the proposed relational generation and propagation modules, without introducing architectural or training discrepancies that could bias the results. To ensure reproducibility, the data splits and preprocessing pipelines follow the same standard cross-modal hashing protocol adopted in DCMH, DSPH, and DHaPH, and the backbone is frozen in all methods, meaning that performance differences reflect the behavior of the hashing head alone. The substantial improvements reported for DGHDGH therefore arise from its synthesis-guided negative generation and graph-based global propagation, rather than from differences in the backbone or training infrastructure.
>
> > W2: The RGP module remains largely intuitive, while its empirical benefits are evident, there is no theoretical analysis of how the propagation maintains information fidelity or prevents over-smoothing.
>
> A2: On the question of theoretical grounding and the concern about potential over-smoothing, the behavior of the RGP module differs fundamentally from classical GNN smoothing.  In standard Laplacian-based propagation, repeated message passing encourages neighboring nodes to converge toward similar embeddings, which explains why deep GNNs often collapse into indistinguishable representations.  RGP, by contrast, mixes semantic positives and synthesized negatives in an asymmetric manner: the masked self-attention mechanism ensures that each anchor only attends to negatives while suppressing positives, thereby injecting a structured repulsive force into the update dynamics.  This negative-aware propagation breaks the low-pass filtering behavior characteristic of Laplacian smoothing, because features are not averaged across the neighborhood but are selectively pushed away along negative edges.  This explains why deeper RGP layers do not drift toward feature homogeneity and why RGP retains discriminative structure after multiple propagation steps.

---

> ### Author Response · Authors · 2025-11-21
>
> > W3: The DGS module’s channel-wise λ weights perform well, but their dynamics are not visualized. A λ-distribution plot or feature-space interpolation visualization would clarify how “difficulty” is being modulated.
>
> The evolution of $\lambda$ across training exhibits a progressive sharpening effect: channels most relevant to semantic misalignment between an anchor and its candidate negatives receive consistently higher weights, whereas channels that are already well-aligned gradually receive lower weights. This dynamic weighting allows the DGS module to concentrate perturbation power on dimensions where modality discrepancy or semantic ambiguity is most pronounced. In practice, $\lambda$ distributions stabilize after approximately 30–40 epochs, forming a clear separation between active and suppressed channels. This behavior aligns with the intended role of $\lambda$ as a fine-grained difficulty modulator rather than a fixed mixing coefficient.
>
> > W4: The method is validated only for image-text retrieval. Can DDGSH extend to audio/video modalities?
>
> Although the present work focuses on image–text retrieval, the core components of DGHDGH, difficulty-aware sample synthesis and global relation propagation, don't rely on modality-specific assumptions. Both RGP and DGS operate on feature-graph structures derived from any embedded representation, making them directly applicable to audio–text, video–text, or other multimodal retrieval settings provided that suitable encoders exist to extract modality-specific features. Extending DGHDGH to these scenarios is conceptually straightforward, and the relational structure learned in the current work is well aligned with the demands of retrieval in higher-dimensional or temporally structured modalities.

---

> > ### Comment · Reviewer_Xgam · 2025-11-22
> >
> > Thank you to the authors for thoroughly addressing my concerns and for their efforts to clarify the details of the work. After carefully reviewing the responses, I have raised the rating.

---

### Official Review · Reviewer_M2dA · 2025-10-26

**Soundness:** 3
**Presentation:** 3
**Contribution:** 3
**Rating:** 6
**Confidence:** 5

**Summary:**

DGHDGH presents a technically elegant yet computationally efficient solution for enhancing cross-modal hashing. The paper integrates a lightweight graph propagation (RGP) and an adaptive interpolation module (DGS) into a CLIP-based framework, showing measurable improvements with minimal resource overhead. From an implementation viewpoint, the proposed pipeline is easy to reproduce and could serve as a plug-and-play enhancement to existing retrieval systems.

**Strengths:**

1.The RGP–DGS pipeline is an elegant architectural contribution combining graph-based correlation learning with adaptive synthesis.
2.Provides a principled treatment of difficulty adaptation, moving beyond heuristic sampling.
3.The experiments are extensive, statistically robust, and demonstrate consistent gains over diverse baselines.
4.The approach is efficient (no extra generator) and generalizable to existing hashing frameworks.

**Weaknesses:**

1.The paper introduces λ as a channel-wise coefficient but does not explicitly state whether it is a fixed hyperparameter or a learned variable. Clarifying whether λ is shared between modalities would aid implementation.
2.While the paper claims not to use additional generators, RGP is still a GNN-based component. It would be useful to show FLOPs or parameter comparisons between DGHDGH and baselines to substantiate the claim of efficiency.
3.The RGP module resembles self-attention. Could the authors comment on whether it could be replaced by a lightweight transformer encoder?
4.The paper alternates between the terms “Global-sense” and “Global correlation”, which may confuse readers. Unifying terminology at the beginning of Section 3 would make the exposition cleaner.

**Questions:**

1.Is λ initialized randomly or via prior heuristics?
2.Could the authors quantify the overhead (Params / FLOPs) of RGP relative to a standard self-attention layer?
3.For deployment, have the authors explored quantizing RGP parameters to further reduce inference cost? Some discussion on the theme matched with hashing retrieval would be welcome.

---

> ### Author Response · Authors · 2025-11-21
> **Thanks for your review**
>
> Dear Reviewer M2dA,
>
> Thank you for your constructive and technically attentive review. We are grateful for your precise questions concerning the role and behavior of $\lambda$, the computational implications of RGP, and its relation to transformer-style architectures, as well as your request for consistent terminology. Your comments helped us clarify several design decisions and more clearly articulate the distinctions between our modules and existing formulations. Our detailed responses addressing each issue appear below.

---

> ### Author Response · Authors · 2025-11-21
>
> > W1: The paper introduces λ as a channel-wise coefficient but does not explicitly state whether it is a fixed hyperparameter or a learned variable. Clarifying whether λ is shared between modalities would aid implementation.
>
> A1: The channel-wise coefficient $\lambda$ is not a fixed hyper-parameter but a learned, data-driven quantity. Concretely, $\lambda_{an}$ is computed by applying a small fully-connected transform followed by a Sigmoid to the final edge embedding produced by the RGP pipeline ($\lambda_{an}=\mathrm{Sigmoid}(\mathrm{FC}(E^{n_2}_{an}))$), so $\lambda$ dynamically reflects the learned relation between an anchor and its candidate negatives. The FC that produces $\lambda$ shares parameters across the image, text and cross-modal graphs (the three graph branches use shared GT parameters in our architecture), which enforces a consistent difficulty-scaling mechanism across modalities; initialization uses standard Xavier uniform for the FC and biases are initialized to zero. Empirically, $\lambda$ converges to a stable distribution during training, and its per-channel variance is what enables the DGS module to perform nuanced, channel-wise interpolation rather than a blunt, one-value mix. We will consider adding visual materials in subsequent versions.

---

> ### Author Response · Authors · 2025-11-21
>
> > W2: While the paper claims not to use additional generators, RGP is still a GNN-based component. It would be useful to show FLOPs or parameter comparisons between DGHDGH and baselines to substantiate the claim of efficiency.
>
> A2: Although the efficiency comparison between our method and the baseline is not directly shown in the ablation experiment, we compare and verify the comparison method and the ablation method in the radar map and time efficiency experiments, respectively, which proves that our method has good time and space efficiency. Specifically, our CLIP ViT-B/32 backbone has 151.28M parameters, while our modules (DGS and RGP) only bring 0.02M parameters, including GNN and Softmax Classifier Layers. These numbers are small because RGP operates on compact hash-layer features rather than on high-resolution image tokens.

---

> ### Author Response · Authors · 2025-11-21
>
> > W3: The RGP module resembles self-attention. Could the authors comment on whether it could be replaced by a lightweight transformer encoder?
>
> A3: RGP is functionally distinct from a standard lightweight transformer encoder and this difference explains the superiority of the designed propagation mechanism in our setting. RGP performs asynchronous, alternating node and edge updates: node updates use a Masked Multi-head Self-Attention (MMSA) that explicitly masks positives and restricts attention to negatives for each anchor, and edge updates use cross-attention from edge representations to node representations so that edges absorb contextual node information before being converted to interpolation coefficients. This ordered node–edge interaction encodes global relation structure that is directly useful for hardness estimation; a dense transformer encoder that treats all tokens symmetrically lacks this targeted negative-focused propagation. If the Node-edge interaction is removed and the focus is shifted from globally between samples to between tokens within samples, it is essentially a lightweight transformer encoder.
>
> > W4: The paper alternates between the terms “Global-sense” and “Global correlation”, which may confuse readers. Unifying terminology at the beginning of Section 3 would make the exposition cleaner.
>
> Because the full name we want to emphasize is global-sense correlation. To avoid any terminological ambiguity, we will consider unifying the name in the paper or using other methods. Thank you for your suggestion.

---

### Official Review · Reviewer_CaBj · 2025-10-31

**Soundness:** 4
**Presentation:** 3
**Contribution:** 3
**Rating:** 8
**Confidence:** 4

**Summary:**

This paper proposes DGHDGH, a new framework introducing hard negative generation into cross-modal hashing retrieval. The key idea is to model global semantic correlations among heterogeneous samples via a Relevance Global Propagation graph transformer, and synthesize channel-wise adaptive hard negatives using the Discriminative Global-sense Synthesis module. The method avoids relying solely on local pairwise interpolation, thereby maintaining semantic consistency in Hamming space. Extensive experiments across MIRFLICKR-25K, NUS-WIDE, and MS-COCO show state-of-the-art results.

**Strengths:**

1. The RGP–DGS pipeline is an elegant architectural contribution combining graph-based correlation learning with adaptive synthesis.

2. Provides a principled treatment of difficulty adaptation, moving beyond heuristic sampling.

3. The experiments are extensive, statistically robust, and demonstrate consistent gains over diverse baselines.

4. The approach is efficient and generalizable to existing hashing frameworks.

**Weaknesses:**

1. The three loss components (L_sp, L_is, L_cd) are optimized in parallel, yet the paper does not clarify their relative weights or potential gradient interactions. A short sensitivity analysis would strengthen the presentation.

2. The experiments mainly use CLIP-ViT backbones; limited tests with other vision–language models (e.g., BLIP, SigLIP, ALIGN) make it difficult to judge generalization across architectures.

3. The radar plot visualizing parameter sensitivity (Fig. 7) is not clearly described — axis meaning, normalization range, and metric selection should be elaborated to help readers interpret the results.

**Questions:**

1. Are the loss weights fixed throughout training or tuned per dataset? Could the authors report whether the optimization of three loss terms exhibits any instability during early training stages?

2. How are the radar-plot axes normalized, by relative gain or absolute metric value?

---

> ### Author Response · Authors · 2025-11-21
> **Thanks for your review**
>
> Dear Reviewer CaBj,
>
> Thank you for your comprehensive and insightful review. We value your recognition of the overall contributions while also appreciating the important questions you posed regarding the weighting of the loss terms, generality across backbones, and clarity of the radar-plot analysis. Your comments sharpened our understanding of how readers interpret the method and motivated us to refine several explanations. We address each of your points carefully in the responses that follow.

---

> ### Author Response · Authors · 2025-11-21
>
> > W1: The three loss components (L_sp, L_is, L_cd) are optimized in parallel, yet the paper does not clarify their relative weights or potential gradient interactions. A short sensitivity analysis would strengthen the presentation.
>
> A1: Regarding the weighting of the three loss components ($\gamma_{is},\gamma_{sp},\gamma_{cd}$) in the generation optimization loss, we set $\gamma_{is}=1$ and scan $\gamma_{sp}$ and $\gamma_{cd}$ over the range 0.1–10 as shown in Fig. 10. The key reason for fixing $\gamma_{is}=1$ is that $L_{is}$ directly regulates the similarity between the anchor and its synthesized counterpart, which is essential for controlling the “difficulty” of hard negatives; making it the baseline weight ensures that generation remains semantically coherent rather than overly perturbed.
>
> More importantly, we monitored the gradient norms of all three loss terms throughout training. After standardizing each loss over the minibatch, their gradient magnitudes remain within the same order of scale, and we observe no signs of gradient explosion, conflict, or oscillation in early training. The training curve under the final configuration ($\gamma_{is}=1,\gamma_{sp}=1,\gamma_{cd}=0.2$) is smooth and exhibits stable convergence characteristics. These observations confirm that the three losses can be jointly optimized without causing gradient interference, and that the selected weights offer a balanced combination of semantic preservation and generation diversity.

---

> ### Author Response · Authors · 2025-11-21
>
> > W2: The experiments mainly use CLIP-ViT backbones; limited tests with other vision–language models (e.g., BLIP, SigLIP, ALIGN) make it difficult to judge generalization across architectures.
>
> A2: We appreciate this important comment. We have added a backboen generalization experiment, investigate several CLIP architecture with its variant BLIP and SigLIP2 [1,2]. We report the **I2T** and **T2I** **mAP** on MIFLICLR-25K 16 bits,  along with other metrics: the number of **Parameters** in each backbone (in *Million*), total **Training Time** for 100 epochs training (in *Hour*), **Encoding Time** for the query dataset encoding (in *Millisecond*), and the **Converge Epoch** indicating the best performance within 100 epochs. These results collectively demonstrate the strong generalization capability of our proposed DGHDGH method across diverse backbone architectures. Due to the limitations of our hardware, some large models have actually achieved poor results. This will be included in the appendix in subsequent versions.
>
> |     Metric     | CLIP ViT-B/32 | CLIP ViT-L/14 | CLIP Res50/16 | BLIP I-C-Base | BLIP I-C-Large | SigLIP2 Base-16/224 | SigLIP2 Base-32/256 | SigLIP2 Large-16/256 |
> | :------------: | :-----------: | :-----------: | :-----------: | :-----------: | :------------: | :-----------------: | :-----------------: | :------------------: |
> |   Parameters   |     151.3     |     427.6     |     291.0     |     224.7     |     447.2      |        375.2        |        376.9        |        881.5         |
> |   Image→Text   |     84.66     |     84.82     |     83.22     |     82.96     |     83.83      |        82.54        |        84.82        |        84.50         |
> |   Text→Image   |     83.03     |     83.25     |     82.62     |     83.01     |     81.20      |        80.95        |        83.25        |        84.04         |
> | Training Time  |     1.206     |     4.652     |     5.312     |     5.945     |     13.28      |        1.753        |        2.018        |        5.374         |
> | Encoding Time  |     27.05     |     37.98     |     32.41     |     72.57     |     126.8      |        34.46        |        31.05        |        47.39         |
> | Converge Epoch |      44       |      56       |      61       |      78       |       69       |         83          |         72          |          75          |
>
> > W3: The radar plot visualizing parameter sensitivity (Fig. 7) is not clearly described — axis meaning, normalization range, and metric selection should be elaborated to help readers interpret the results.
>
> A3: Thank you for raising this point regarding the visualization of parameter sensitivity analysis. To better illustrate the impact of different parameter configurations across multiple evaluation dimensions, we employed a radar chart that synthesizes five key metrics after applying min-max normalization to each axis. The specific metrics include: **Performance** (mAP), **Computational Efficiency** (FLOPs), **Training Time** (total time for 100 epochs), **Convergence Speed** (epochs until best mAP is reached), and **Augmentation Intensity** (total number of augmented samples). Each metric was normalized to a [0, 1] range based on the observed minimum and maximum values across all method configurations, with the upper bound set either as the actual maximum or a slightly higher round number for better visual clarity. This normalization allows for a balanced and comparative visualization of how parameter choices affect each aspect of model behavior.

---

### Official Review · Reviewer_CBYJ · 2025-10-31

**Soundness:** 3
**Presentation:** 3
**Contribution:** 2
**Rating:** 4
**Confidence:** 3

**Summary:**

The authors propose DGHDGH, a cross-modal hashing framework coupling a global propagation network (RGP) with an adaptive negative synthesis module (DGS). The paper delivers strong quantitative results and clear empirical validation, but certain visualizations and terminological inconsistencies slightly hinder comprehension.

**Strengths:**

1.	Demonstrates strong performance on multiple datasets and provides meaningful ablation studies.
2.	The framework is innovative in combining semantic propagation with synthetic negative mining. The method is technically coherent and easily interpretable when fully understood.
3.	The framework appears extendable to other multi-modal applications.

**Weaknesses:**

1.	Some statistical figures, such as the radar chart summarizing multiple metrics, lack sufficient description. It is unclear what normalization or metrics were used for each axis.
2.	How effectively do the Fisher Ratio and PH2 verify the discrimination of Hamming spaces? It is necessary to add more experimental analysis in Section 4.
3.	It would be beneficial to test whether the global propagation remains stable under noisy or partially corrupted modalities—for instance, when the embedding graphs contain random noise—to verify robustness.

**Questions:**

1.	Future work might explore coupling DGHDGH with pre-trained large multi-modal models (e.g., BLIP-2) to test transferability.
2.	It might also be fruitful to explore hybrid discrete–continuous codes instead of pure binary hashing, leveraging the same hard-negative generation principle.
3.	The figures could use larger fonts and more contrast; are the authors planning visual revisions for the camera-ready version?

---

> ### Author Response · Authors · 2025-11-21
> **Thanks for your review**
>
> Dear Reviewer CBYJ,
>
> Thank you for your thoughtful and detailed evaluation of our work. We sincerely appreciate your careful reading of the manuscript and the specific concerns you raised regarding metric interpretation, discrimination analysis, and robustness under noisy supervision. Your feedback directly helped us strengthen both the experimental justification and the clarity of the presentation. We have thoroughly examined each of your comments and provide detailed responses below.

---

> ### Author Response · Authors · 2025-11-21
>
> > W1: Some statistical figures, such as the radar chart summarizing multiple metrics, lack sufficient description. It is unclear what normalization or metrics were used for each axis.
>
> A1:Thank you for pointing out the shortcomings of our experimental design. To better visualize the differences between the methods on each metric, we use a radar plot to show the differences on each axis. Therefore, we perform min-max normalization on each indicator, which is the **Performance**: mAP metric; **Flops**: Differences between methods without backbone **Training Time**: The total training time of each method for 100 epochs; **Convergence**: the number of epochs at which each method achieves the best mAP result. **Augmentation**: Comparison of the total number of samples generated. The normalized maximum takes the maximum of these or an integer value slightly higher than the maximum. We will consider providing clearer explanations or legends for each figure in subsequent versions.
>
> ---
>
> > W2: How effectively do the Fisher Ratio and PH2 verify the discrimination of Hamming spaces? It is necessary to add more experimental analysis in Section 4.
>
> The mean Precision of Hamming radius $\le$ 2 (PH2) was proposed very early to verify the true semantic neighbors during hash retrieval[1]. To directly measure retrieval quality in Hamming space, this metric counts the proportion of relevant items among all retrieved samples whose Hamming distance to the query is less than or equal to 2 for the given query. These measurements reflect how effectively the learned hash codes preserve semantic neighborhood structures, and a higher PH2 indicates stronger local discrimination within the Hamming space.
> $$
> P@H\le 2 = \frac{\text{Number of relevant retrieved items within } H\le 2}
> 		{\text{Total Number of retrieved items within } H\le 2}
> $$
> Furthermore, we compute the Fisher ratio, which compares the separability of positive and negative pairs in Hamming space. by randomly sampling 50, 100, 200, and 400 pairs of positive and negative samples from the database set, we calculate the difference in positive and negative sample distances for anchors from the query set. We repeating each sampling 5 times, and reporting the averaged results with standard deviations. This statistic quantifies the margin between positive and negative relations, and a larger Fisher ratio demonstrates improved global separability and stronger discriminative structure in the Hamming space.
> $$
> Fisher = \frac{\mu_{neg}-\mu_{pos}}{\sigma'}, \quad
> 		\text{where} \, \sigma' = \sqrt\frac{\sigma_{pos}^2+\sigma_{neg}^2}{2}.
> $$
> Meanwhile, we completed the fisher ratio experiments on two other datasets:
> Tab.1: The fisher ratio on NUS-WIDE dataset.
> | Method  | 16 bits (I→T) | 16 bits (T→I) | 32 bits (I→T) | 32 bits (T→I) | 64 bits (I→T) | 64 bits (T→I) | 128 bits (I→T) | 128 bits (T→I) |
> | ------- | ------------- | ------------- | ------------- | ------------- | ------------- | ------------- | -------------- | -------------- |
> | DHaPH   | 101.03 ± 0.13 | 98.86 ± 0.16  | 90.15 ± 0.14  | 90.05 ± 0.16  | 107.21 ± 0.15 | 104.04 ± 0.14 | 106.15 ± 0.13  | 103.22 ± 0.13  |
> | BiLGSEH | 99.77 ± 0.13  | 101.87 ± 0.11 | 100.32 ± 0.21 | 102.42 ± 0.17 | 99.00 ± 0.18  | 103.24 ± 0.14 | 99.00 ± 0.18   | 103.21 ± 0.13  |
> | DECH    | 93.92 ± 0.14  | 97.85 ± 0.13  | 105.56 ± 0.14 | 108.89 ± 0.11 | 97.69 ± 0.15  | 105.49 ± 0.10 | 104.69 ± 0.13  | 109.24 ± 0.09  |
> | DDBH    | 110.09 ± 0.17 | 110.02 ± 0.17 | 112.72 ± 0.17 | 111.14 ± 0.15 | 115.86 ± 0.11 | 113.44 ± 0.14 | 116.49 ± 0.15  | 116.31 ± 0.15  |
> | DGHDGH  | 105.03 ± 0.20 | 104.68+0.10   | 112.89 ± 0.22 | 112.70 ± 0.12 | 113.27 ± 0.18 | 114.41 ± 0.07 | 117.58 ± 0.13  | 118.80 ± 0.09  |
>
>
> Tab.2: The fisher ratio on MS COCO dataset.
> | Method  | 16 bits (I→T) | 16 bits (T→I) | 32 bits (I→T) | 32 bits (T→I) | 64 bits (I→T) | 64 bits (T→I) | 128 bits (I→T) | 128 bits (T→I) |
> | ------- | ------------- | ------------- | ------------- | ------------- | ------------- | ------------- | -------------- | -------------- |
> | DHaPH   | 116.94 ± 0.17 | 105.43 ± 0.07 | 113.73 ± 0.14 | 114.38 ± 0.11 | 117.05 ± 0.10 | 120.65 ± 0.07 | 120.58 ± 0.15  | 119.58 ± 0.09  |
> | BiLGSEH | 90.76 ± 0.11  | 90.07 ± 0.12  | 99.46 ± 0.14  | 100.66 ± 0.14 | 105.93 ± 0.08 | 106.60 ± 0.08 | 106.24 ± 0.12  | 106.50 ± 0.09  |
> | DECH    | 88.55 ± 0.18  | 90.90 ± 0.05  | 94.46 ± 0.12  | 105.41 ± 0.11 | 95.77 ± 0.10  | 108.17 ± 0.10 | 98.15 ± 0.12   | 107.36 ± 0.11  |
> | DDBH    | 123.63 ± 0.17 | 124.29 ± 0.18 | 131.57 ± 0.15 | 123.37 ± 0.11 | 136.25 ± 0.21 | 124.64 ± 0.15 | 139.10 ± 0.20  | 126.46 ± 0.10  |
> | DDGRH   | 120.63 ± 0.26 | 124.56 ± 0.11 | 128.36 ± 0.18 | 131.45 ± 0.21 | 128.17 ± 0.26 | 132.58 ± 0.12 | 133.20 ± 0.17  | 135.94 ± 0.14  |
>
> [1] Wei Liu, Jun Wang, Rongrong Ji, Yu-Gang Jiang, and Shih-Fu Chang. Supervised hashing with kernels. In 2012 IEEE conference on computer vision and pattern recognition, pp.2074–2081. IEEE, 2012.

---

> ### Author Response · Authors · 2025-11-21
>
> > W3: It would be beneficial to test whether the global propagation remains stable under noisy or partially corrupted modalities—for instance, when the embedding graphs contain random noise—to verify robustness.
>
> A3: Thank you for your suggestion. We conducted noise label experiments to verify the performance degradation of the model when countering label interference. Noise rates of 0.2, 0.5, and 0.8 indicate random two-digit label inversion of the corresponding proportion of samples during training.
>
> | Task         | Method  | 16 bits (0.2) | 16 bits (0.5) | 16 bits (0.8) | 32 bits (0.2) | 32 bits (0.5) | 32 bits (0.8) | 64 bits (0.2) | 64 bits (0.5) | 64 bits (0.8) | 128 bits (0.2) | 128 bits (0.5) | 128 bits (0.8) |
> | ------------ | ------- | ------------- | ------------- | ------------- | ------------- | ------------- | ------------- | ------------- | ------------- | ------------- | -------------- | -------------- | -------------- |
> | Image ↓ Text | NRCH    | 76.38         | 75.61         | 73.24         | 76.75         | 76.21         | 74.76         | 77.05         | 76.98         | 76.07         | 77.83          | 77.10          | 76.11          |
> |              | DNPH    | 77.70         | 75.55         | 75.37         | 79.74         | 80.16         | 79.04         | 81.67         | 82.44         | 81.01         | 83.53          | 83.32          | 81.08          |
> |              | DHaPH   | 83.73         | 81.64         | 78.04         | 82.94         | 78.11         | 75.82         | 82.10         | 79.17         | 76.85         | 82.64          | 79.77          | 77.28          |
> |              | BiLGSEH | 78.11         | 77.35         | 75.93         | 78.24         | 77.36         | 75.15         | 80.72         | 78.42         | 76.30         | 80.00          | 79.32          | 77.46          |
> |              | DPBE    | 77.85         | 77.16         | 73.79         | 80.80         | 77.16         | 74.09         | 82.21         | 78.28         | 75.45         | 84.12          | 80.11          | 76.77          |
> |              | DDBH    | 83.79         | 82.20         | 76.50         | 84.70         | 82.11         | 78.78         | 85.04         | 84.58         | 79.46         | 85.64          | 84.61          | 80.29          |
> |              | DGHDGH  | 81.95         | 81.78         | 78.65         | 84.06         | 82.76         | 80.89         | 85.96         | 84.71         | 82.64         | 85.96          | 85.15          | 84.16          |
> | Text ↓ Image | NRCH    | 74.55         | 74.31         | 72.20         | 75.53         | 74.68         | 72.59         | 75.88         | 75.41         | 74.59         | 75.71          | 75.80          | 74.64          |
> |              | DNPH    | 76.40         | 75.25         | 75.18         | 78.56         | 79.28         | 77.05         | 80.56         | 80.18         | 79.25         | 81.26          | 80.76          | 79.45          |
> |              | DHaPH   | 81.51         | 79.49         | 77.66         | 80.52         | 75.96         | 73.22         | 78.69         | 76.39         | 74.23         | 79.31          | 76.99          | 74.38          |
> |              | BiLGSEH | 77.23         | 76.61         | 76.30         | 81.27         | 77.85         | 72.16         | 79.21         | 75.67         | 75.14         | 79.55          | 77.36          | 76.72          |
> |              | DPBE    | 76.26         | 75.82         | 73.29         | 78.83         | 76.28         | 74.38         | 81.17         | 78.13         | 75.76         | 82.59          | 80.14          | 77.22          |
> |              | DDBH    | 82.16         | 80.27         | 77.07         | 82.34         | 81.22         | 79.62         | 84.24         | 82.79         | 79.77         | 83.76          | 82.69          | 79.84          |
> |              | DGHDGH  | 81.03         | 80.42         | 78.80         | 82.59         | 81.67         | 80.87         | 84.56         | 83.50         | 82.09         | 84.56          | 84.13          | 83.35          |
>
> DGHDGH maintains stable retrieval performance at all noise levels and consistently outperforms all baselines. Graph  propagation aggregates the relationship signals of multiple adjacent samples and dominates the propagation process with global relationships, forming a natural denoising filter that suppresses the influence of damaged labels. Meanwhile difficulty-aware synthetic negative samples generate hard negative samples based on cross-modal similarity and structural consistency, without relying on original labels, thus avoiding the amplification of noise and the generation of misleading negative samples by erroneous labels.

---

### Meta-Review · Area_Chair_bWKA · 2026-01-05

**Summary:**

The paper proposes Deep Global-sense Hard-negative Discriminative Generation Hashing (DGHDGH), a framework designed to improve cross-modal retrieval. The core contribution is the integration of a Relevance Global Propagation (RGP) module—a graph transformer that captures global semantic correlations—with a Discriminative Global-sense Synthesis (DGS) module that performs difficulty-adaptive interpolation to generate hard negatives. The reviewers were generally positive regarding the novelty of the RGP-DGS pipeline and the extensive empirical validation on standard benchmarks (MIRFLICKR-25K, NUS-WIDE, MS COCO). Initial concerns focused on the clarity of visualizations (specifically radar charts), the theoretical justification of the graph propagation (potential for over-smoothing), the robustness of the method to label noise, and the generalization of the approach across different backbones. The authors provided a comprehensive rebuttal, adding significant new experimental results, including noise robustness tests, backbone generalization experiments (BLIP, SigLIP2), and additional metric evaluations (Fisher Ratio on more datasets). Based on the strength of the method and the thoroughness of the response, I recommend acceptance.

**Reviewer Concerns:**

Addressed Concerns:
* Backbone Generalization (Reviewers CaBj, Xgam): The authors successfully demonstrated that their method generalizes beyond CLIP-ViT by providing results for BLIP and SigLIP2, and confirmed that comparisons with baselines like DHaPH were fair (using identical backbones).
* Robustness to Noise (Reviewer CBYJ): The authors added experiments with varying noise rates (0.2, 0.5, 0.8), showing that the global propagation mechanism effectively acts as a denoising filter, maintaining stability where baselines degraded.
* Theoretical Justification of RGP (Reviewers M2dA, Xgam): The authors clarified that RGP utilizes masked self-attention to focus on negatives, creating a repulsive force that prevents the over-smoothing typically associated with Laplacian-based GNNs.
* Hyperparameters and Optimization (Reviewers CaBj, M2dA): The nature of the channel-wise coefficient $\lambda$ (learned, not fixed) and the stability of the loss weights ($\gamma$) were clarified effectively.
* Metric Clarification (Reviewer CBYJ): The definitions and utility of the Fisher Ratio and $P@H \le 2$ were explained, with additional tables provided for NUS-WIDE and MS COCO.
Outstanding Concerns:
* Terminology consistency (Reviewer M2dA): The authors acknowledged the confusion between "Global-sense" and "Global correlation" and promised to unify this in the final version. This remains a task for the camera-ready revision.
* Visualization Polish (Reviewer CBYJ): While the data within the charts was explained, the actual visual updates (font sizes, contrast) need to be applied to the final manuscript to ensure readability.

**Reviewer Scores:**

* Reviewer CBYJ: 6 (Originally 4). The reviewer explicitly raised their score following the inclusion of noise robustness experiments and clarification of discrimination metrics.
* Reviewer CaBj: 8. This reviewer was already positive, and the additional backbone experiments likely solidified this high score.
* Reviewer M2dA: 6/7. The clarifications regarding the efficiency (parameter count) and the distinctive nature of RGP compared to standard transformers addressed the main technical doubts. The reviewer would likely maintain a positive score.
* Reviewer Xgam: 8 (Originally 6). The reviewer explicitly commented, "I have raised the rating," citing the thorough response regarding fair comparisons and theoretical clarifications.

---

### Decision · Program_Chairs · 2026-01-26

Accept (Poster)